# Balancing Agriculture and Industry through Waste Utilization for Sugarcane Sustainability

Arika Bridhikitti [1,2,*], Jutamas Kaewsuk [1], Netiya Karaket [3], Richard Friend [4], Brett Sallach [4], James P. J. Chong [5] and Kelly R. Redeker [5]

1   Environmental Engineering and Disaster Management Program, School of Interdisciplinary Studies, Mahidol University Kanchanaburi Campus, Kanchanaburi 71150, Thailand; jutamas.kae@mahidol.edu
2   Earth Science Research Center, Mahidol University Kanchanaburi Campus, Kanchanaburi 71150, Thailand
3   Agricultural Science Program, School of Interdisciplinary Studies, Mahidol University Kanchanaburi Campus, Kanchanaburi 71150, Thailand; netiya.kar@mahidol.edu
4   Department of Environment and Geography, University of York, Heslington, York YO10 5DD, UK; richard.friend@york.ac.uk (R.F.); brett.sallach@york.ac.uk (B.S.)
5   Department of Biology, University of York, Heslington, York YO10 5DD, UK; james.chong@york.ac.uk (J.P.J.C.); kelly.redeker@york.ac.uk (K.R.R.)
*   Correspondence: arika.bri@mahidol.edu

**Abstract:** The Bio-Circular-Green Economy initiative adopted in Thailand encourages using sugar mill by-products for food production, benefiting both farmers and the environment. This study assesses the feasibility of applying by-products from the sugar mills and distilleries into sugar plantations for irrigation, fertilization, and soil conditioning. It addresses challenges in sustainable waste utilization and offers recommendations. This study reviews literature, conducts preliminary surveys, and analyzes samples from sugarcane fields in Kanchanaburi, Thailand. The findings reveal that while vinasse and filter cake demonstrate promise as soil conditioners, their application requires careful consideration of soil type and pre-treatment processes. Vinasse, rich in essential nutrients, can benefit sandy soils by improving microbial activity and nutrient availability. Filter cake exhibits positive effects on soil texture, water permeability, and mineral content. Treated wastewater can be used for ferti-irrigation. However, about one-third of farmers lack confidence in its use due to concerns about limited nutrient availability, high transportation costs, and potential negative impacts on health, agriculture, and the ecosystem. To enhance farmer adaptability and ensure the successful utilization of waste, several challenges must be addressed, including: (1) assessing financial and technical feasibility of waste transportation and value-added products, (2) overcoming regulatory barriers related to transportation and utilization of industrial wastes, (3) disseminating knowledge to farmers regarding proper fertilization and waste utilization practices, and (4) implementing long-term monitoring on ecosystem health and conducting sustainability assessments of the waste utilization to affirm sustainability attainment.

**Keywords:** Bio-Circular-Green Economy; ethanol; sugar; vinasse; waste utilization; sustainable agriculture



## 1. Introduction

Sustainability has been in the limelight since 2015, when the 2030 Agenda for Sustainable Development, with 17 sustainability goals, was adopted by all United Nations Member States. These goals promote responsible production and consumption, quality employment, environmental health, and adaptability to climate change. To drive the goals, many countries adopted concepts of Circular Economy, Green Economy, and/or Bioeconomy, which all share the common ideal of integrating economic, environmental, and social developments [1]. The Circular Economy and Bioeconomy are resource-based. The main aspects of the Circular Economy are reuse/recycling, eco-efficiency, industrial symbiosis,

and greener supply chains, whereas the Bioeconomy is more focused on the biosecurity of crops, species, and yield. The Green Economy plays the key underpinning role of all ecological processes (water, land, biodiversity, and food security) and is more focused at the local level, targeting eco-tourism, business, education [1], etc. To promote economic growth, the Thai Government has introduced a Bio-Circular-Green Economy (or BCG). The government aims to raise national Gross Domestic Product (GDP) and employment numbers in four industrial sectors targeted by BCG, which are as follows: 1. Food and Agriculture, 2. Medical and Wellness, 3 Energy, Material and Biochemical, and 4. Tourism and Creative Economy [2].

Thailand's sugar industry is vital to the country's economy. Currently, Thailand is the world's second largest sugar exporter after Brazil, accounting for 19% of total global exported sugar [3]. The sugarcane cultivation area in Thailand covers 1.92 M ha (year 2019) and provides resource to 57 sugar mills (year 2021). In the production year 2020/21, there was total national cane yield of 74.9 M ton with total sugar production of over 8.3 M ton year$^{-1}$. This cane yield was 41.7% less than average yields due to drought [3]. The sugar processing can yield not only sucrose products, but also other food, feed, biofertilizers, bioplastics, and biomolecules [4]. It has been found in many parts of the world that diversification of sugar mill outputs could improve the sugarcane value chain, ensure eco-efficiency of sugar production, and contribute to sustainable goals [4].

Once loaded, sugarcanes are cleaned with water and later separated into juice and wood pulp (or bagasse) through milling. The raw juice is sieved and clarified under pressure, which results in filter cake as a by-product. Several chemicals are added during the clarification stage, mainly for coagulation of impurities, pH adjustment of juice, and sludge stabilization [5]. The clear juice is later vaporized, crystallized, and centrifuged to separate sugar crystals and cooked mass, or molasses. The molasses can be further used as an input for ethanol production. This process produces vinasse as wastewater. In summary, the by-products and wastes from the sugar–ethanol industry include bagasse (298 kg ton$^{-1}$ cane), filter cake (30 kg ton$^{-1}$ cane), molasses (34 kg ton$^{-1}$ cane), wastewater (70 kg ton$^{-1}$ cane) from the sugar mill industry, and vinasse (13 L L$^{-1}$ ethanol produced or 3376 L ton$^{-1}$ molasses consumed) from the ethanol industry [6–8]. Wastewater comes from various stages, includes cane cleaning, air treatment, cooling, condensed effluent, barometric condensers, boiler blowdown, floor/machine cleaning, and vinasse (from ethanol production) [7].

Zero waste production is highly possible for the new era of sugar industries. Some by-products from sugar mills are used in energy production, particularly bagasse and molasses. About 3885 G Wh electricity or about 2% of total electricity production in Thailand was derived from bagasse in 2016 [9,10]. Molasses, with a current production of ~1.9 M L d$^{-1}$ [9], is typically used for ethanol production, following the processes of fermentation and distillation. The demand for molasses-based ethanol is rising in Thailand due to energy security concerns and eco-efficiency demands. Nguyen et al. [11] showed a positive net energy of replacing fossil fuels with the molasses-based ethanol in Thailand. In response to the benefit of increased bioethanol, the Department of Alternative Energy Development and Energy Conservation [12] aims to increase the total sugarcane plantation area by ~1.6 times and double sugarcane yield by 2026. Other by-products or waste from sugar mills and ethanol industries are intended for use in other processes, including food production.

Gonçales Filho et al. [7] also conducted studies of production within Circular/Green Economy in the sugar–ethanol industry in Brazil, and recommended recycling sugar mill by-products, including vinasse, filter cake, and boiler ash, as biofertilizers. Life cycle assessment confirmed the advantage of utilizing sugar mill waste for alcohol production, biogas, animal foods, and fertilizers to mitigate environmental and health impacts [6]. Thus, applying organic wastes from sugar mills and molasses-based distilleries to sugarcane fields could be not only beneficial for farmers' livelihood, but also improve environmental health, leading to better public quality of life as a whole.

While the by-products from the sugar–ethanol industries can be used for agriculture, there are potential negative impacts associated with its utilization. The impacts include soil salinity [13,14], disturbance in soil pH [15], groundwater contamination [16], the accumulation of heavy metals in soils [17], the inhibition of growth in terrestrial plants and aquatic organisms [18], alterations in soil microorganisms [19], and problems associated with regulatory compliance. In Thailand, the recent life cycle assessment shows that using these by-products as soil conditioners and biofertilizers could significantly reduce environmental impacts [20]. Nonetheless, the study on the current practices, farmer perception, and barriers to achieving the BCG goal has not yet been implemented. This study aims to assess the current situation and feasibility of applying the by-products from sugar mills and distilleries back into the sugar plantations for purposes of irrigation, fertilization, and soil conditioning. This study also examines the challenges and offers practical implications for sustainable waste utilization. This information may benefit policy makers to produce strategies leading toward a more successful Bio-Circular-Green Economy. This study is based on reviewing literature worldwide, together with preliminary surveying and analysis of soil and biomass samples taken from fields in Kanchanaburi, Thailand. This work provides valuable insights for farmers seeking sustainable alternatives to chemical fertilizers, as well as for policy makers and researchers interested in advancing the BCG initiative within the sugar–ethanol industry in Thailand.

## 2. Methodology

### 2.1. Description of the Study Area

Kanchanaburi province in the west of Thailand is characterized by mountainous terrain with limestone dominance. Deciduous forest accounts for 61.6% [21] of the land cover in this province, densely located in the west, whereas agricultural lands, mainly cassava and sugarcane, are predominant in the lowlands of the eastern region. The province has the third largest sugarcane plantation area in the country (6.4% of the total national planted area) and has the highest number of sugar mills (8 factories) [3]. The provincial economy is driven by the agricultural sector, especially sugarcane, and the cane price highly influences provincial gross domestic product [21]. The rainy season persists from May to October, with average annual rainfall of 1144.1 mm [21]. Maximum annual temperature ranges from 39.3 to 42.7 °C in the summertime (from March to April) while annual minimum temperatures range from 10.6 to 16.4 °C in the wintertime (from November to January) [22].

### 2.2. In-Depth Interviews and Field Sampling

In-depth interviews and field sampling were carried out from 20 to 31 May 2022. A total of 207 farmers in four sub-districts participated in the interviews. The four sub-districts were Muang Kanchanaburi, Tha Muang, Tha Maka, and Bo Phloi. Overall, farmers in the interviews were mainly male (86.9%), with 89.2% of their croplands being sugarcane plantations. The average plantation area was 38 ha per household, ranging from 0.5 to 384 ha. In Thai culture, the male is responsible for field work, whereas the female takes care of household chores and children. Some females help in their husband's work, but they were unable to provide precise information on field supplies and practices, such as fertilization. Interviews consisted of four sections: (1) farmer information, (2) cane yield and cultivating practices (cropping area, cropping calendar, fertilization, chemical applications, and water sources), (3) adaptions to drought and flood, and (4) attitudes toward utilizing sugar mills' water (confidence level, water characteristics, impacts on health, agriculture, and environment). The interviews complied with the guideline of the Human Research Ethics Committee, Mahidol University (Protocol Number MU-CIRB 2022/038.2302 since 28 April 2022). Thus, no photographs were taken during the interviews, and consent from the interviewees was obtained prior to commencement.

Surface soil (0–15 cm depth) and cane leaf samples were taken from 127 sugarcane plantations owned by 121 out of the 207 interviewed farmers. The sampled leaf was the third top leaf. The sampling locations were not limited to the four districts. Soils were

randomly sampled in zigzag patterns across each field, and field samples were composited from seven to ten sub-field sample sites, depending on plantation size.

Water quality samples were taken from local industries in Bo Phloi and Tha Maka sub-districts. The water samples included treated wastewater from one sugar mill (Plant A) and treated wastewater and vinasse from one sugar–ethanol industry (Plant B). In addition, irrigation water from the nearby agricultural area in Tha Maka was collected twice in the dry season of March and April 2022. Water samples were analyzed for (1) pH, using a portable pH meter (EUTECH pH700 model), (2) dissolved oxygen (DO), using a portable DO analyzer (EUTECH PD650 model), (3) turbidity, using a turbidimeter (EUTECH NT-100), (4) electroconductivity (EC), using a portable EC meter (EUTECH CON11 model), (5) color, using a portable photometer (HANNA HI96727 model), and (6) total nitrogen (TN), total organic carbon (TOC), and inorganic carbon (IC) contents, using Analytikjena analyzer Multi N/C 3100 (quality of replicates of 0.99984 and correlation coefficient of 0.99992). Further analysis of metal content was conducted on the vinasse sample. The levels of metals, including micronutrients (Iron-Fe, Manganese-Mn, Zinc-Zn, and Copper-Cu) and toxic metals (Chromium-Cr, Lead-Pb, Cadmium-Cd, Mercury-Hg, and Arsenic-As) were assessed using Inductively Coupled Plasma-Atomic Emission Spectroscopy.

### 2.3. Sample Preparation and Analysis

Soil samples were air-dried immediately after being sampled and took 1 to 3 days to be completely dried. The dried soils were ground and sieved using 2 mm mesh size. The prepared soil samples were later used for physiochemical soil determination, including soil color (via Munsell Color Book), soil texture (using ribbon test), soil pH (1:1 soil: water), and soil electroconductivity, EC, (1:5 soil: water).

### 2.4. Data Collection

To understand the impacts of farmers on climate change, meteorological parameters and cane inputs to the sugar mills were collected. Monthly rainfall, relative humidity, evaporation, and daily minimum and daily maximum temperatures for the years 2002 to 2020 were acquired from the Thai Meteorological Department. The annual cane inputs (fresh cane and burnt cane) were collected from the Office of the Cane and Sugar Board via an online database, available at http://www.ocsb.go.th/th/cms/detail.php?ID=142&SystemModuleKey=production (accessed on 26 February 2023).

### 3. Results and Discussion

This study concentrates on wastes and by-products derived from sugar mills and molasses-based ethanol industries that are applicable to agricultural production. They include treated wastewater and filter cake from sugar mills and vinasse from the ethanol industries [23]. The results and discussion are categorized into six sections:

- First, an analysis and review of nutrient contents and substances in the by-products and wastes from the sugar mill and distillery industry.
- Second, a discussion of the potential benefits and impacts associated with the utilization of these by-products and wastes as biofertilizers.
- Third, suggestions for the possible application of these by-products and wastes as soil conditioners.
- Fourth, an examination of the requirements and potential for using treated wastewater from the sugar mill and distillery industry for ferti-irrigation.
- Fifth, an exploration of the challenges associated with the utilization of wastes from the sugar mills and ethanol distillery industries, illustrated through the case of Kanchanaburi, Thailand.
- Sixth, a discussion of the limitations of the research implementation.

### 3.1. Nutrient Contents and Substances in the By-Products and Wastes

3.1.1. Treated Wastewater

The wastewater from sugar mills is generally from washing floors, the boiling house, the evaporator, and periodic cleaning using acid or lime water [24]. Therefore, sugar mills' wastewater contains either acidic or alkaline compounds, a considerable concentration of suspended solids and sugar with a high biochemical oxygen demand (BOD) and chemical oxygen demand (COD) [25]. The effluent may also contain harmful substances, including salts and heavy metals (such as Zn, Cu, Fe, Mn, and Pb). Long-term use of waste effluent may cause detrimental effects to plants [17]. Typically, wastewater from sugar mills is treated using a series of naturally aerated ponds, in which nutrients and colors attributed to recalcitrant organic compounds are not effectively removed. The wastewater, therefore, is often a light yellowish-green color.

Based on the pH, EC, and DO (seen in Table 1), the treated wastewaters from the two sugar mills were compliant with the water quality discharge standards into irrigation waterways, issued by the Royal Thai Irrigation Department [26] (pH of 6.5 to 8.5, EC < 2000 $\mu$S cm$^{-1}$, DO > 2 mg L$^{-1}$). Furthermore, a TN content of 4.42 and 0.69 mg L$^{-1}$ (Plant A and Plant B, respectively) for the treated wastewaters suggested that the nitrogen content in the water met the required standards (Total Kjeldahl Nitrogen, TKN < 35 mg L$^{-1}$).

**Table 1.** Physiochemical properties of treated wastewater and vinasse acquired from local industrial plants (Plant A and Plant B) and surface runoff near Plant A.

| | Treated Sugar Mill's Wastewater | Treated Sugar Mill's Wastewater | Anaerobically Treated Vinasse from Molasses-Based Ethanol Production | Surface Runoff in the Agricultural Area Near Plant A | Standard of Water Quality Discharged into the IrrigationWaterway |
|---|---|---|---|---|---|
| Plant | A | B | B | | |
| pH | 8.48 | 7.68 | 8.06 | 7.27–7.49 | 6.5–8.5 |
| Dissolved oxygen, DO (mg L$^{-1}$) | 4.48 | 5.03 | 1.10 | 5.55–7.56 | >2 |
| Turbidity (NTU) | 48.2 | 23.9 | 158 | 4.61–5.75 | NR |
| Electroconductivity, EC ($\mu$S cm$^{-1}$) | 987 | 19.7 | 6140 | 84.7–661 | <2000 |
| Color (PCU) | 320 | 10 | 16,000 | 110 | NR |
| Total Nitrogen, TN (mg L$^{-1}$) | 4.42 | 0.69 | 808.1 | 0.43 | TKN < 35 |
| Total Organic Carbon, TOC (mg L$^{-1}$) | 23.73 | 9.42 | 9040 | 9.66 | NR |
| Inorganic Carbon, IC (mg L$^{-1}$) | 125 | 21.57 | 1440 | 24.93 | NR |
| C/N ratio | 33.65 | 44.91 | 12.97 | 80.44 | NR |

Note: NR = Not reported; TKN = Total Kjeldahl Nitrogen.

3.1.2. Vinasse (Stillage, Spent Wash)

About 80% or higher of sugar mills globally produce both raw sugar and ethanol, and wastewater from the molasses-based ethanol distillery is called vinasse, stillage, or spent wash [8]. Vinasse is typically treated via anaerobic digestion [8,27]. Nonetheless, high organic matter and nutrient loads can be found after treatment [24]. As observed in the treated vinasse in this study, the water exhibited low DO (1.1 mg L$^{-1}$), high EC (6.14 mS cm$^{-1}$), and high TN (808.1 mg L$^{-1}$), which failed to meet the water quality standards regarding discharging into irrigation waterways [26]. The vinasse also contained a high carbon content, including organic carbon of 9040 mg L$^{-1}$ and inorganic carbon of 1440 mg L [1] (see Table 1). The C:N ratio was about 13:1, which is lower than the suggested ratio for efficient biogas production via anaerobic digestion of 20–30:1 [28].

The treated vinasse exhibited a pungent smell and dark brown color. The dark brown color is due to the presence of melanoidin from cane molasses, which is not metabolized effectively by bacteria in typical wastewater treatment processes [8,29]. Previous studies consistently show that raw vinasse is acidic in nature [30], but post-anaerobic digestion, the vinasse may become more neutral [29], as observed in this study (see Table 1).

Vinasse is also a source of Potassium (K), Calcium (Ca), and Magnesium (Mg). Thus, it could reduce the need for inorganic K fertilizer [31]. The considerable amount of Ca and Mg in vinasse could be beneficial for the reclamation of sodic soils, but the vinasse, having high salts, could alternately increase soil salinity [13]. Rajkishore and Vignesh [29] also pointed out that the vinasse could also be a significant source of micronutrients, such as Fe, Mn, Zn, and Cu. However, the magnitudes could exhibit significant variation. Our findings for the post-methanated vinasse indicate that the Fe and Mn could be considerable (19.05 and 8.91 mg kg$^{-1}$, respectively), but Zn (0.61 mg kg$^{-1}$) and Cu (undetectable) were notably lower compared to the vinasses studied by Rajkishore and Vignesh [29], as well as lower than the quantities typically found in organic fertilizers, such as manures and mushroom substrate, reported by Kuziemska et al. [32].

The levels of toxic metals in the post-methanated vinasse were considerably low: 0.04 mg Cr kg$^{-1}$, 0.08 mg Pb kg$^{-1}$, 0.27 mg As kg$^{-1}$, <0.001 mg Hg kg$^{-1}$, and non-detectable Cd. These levels are well below the threshold levels outlined in the national guideline for liquid biofertilizer (5 mg Cr kg$^{-1}$, 3 mg Pb kg$^{-1}$, 3 mg As kg$^{-1}$, 0.5 mg Hg kg$^{-1}$, and 2 mg Cd kg$^{-1}$), as established by the Land Development Department [33].

### 3.1.3. Filter Cake (or Press Mud)

Filter cake is obtained during the purification of sugar by sulfitation and carbonation processes, resulting in high concentrations of sulfate and carbonate [30]. Filter cake is soft, spongy, and dark brown, containing moisture (50–65%), crude wax (7–15%), fiber (20–30%), crude protein (5–10%), sugar (5–12%), and nitrogen (2–2.5%) [15]. Due to the high complexity of the cellulose matrix and lignin, filter cake decomposition requires the aid of various microbes, fungi, and bacteria, particularly actinomycetes [30]. Filter cake contains both macronutrients (C, N, P, K, Mg, Ca, S) and micronutrients (Fe, Mn, and Silica-Si) [15].

### 3.2. Feasibility of Using the By-Products and Wastes as Biofertilizers

### 3.2.1. Current Fertilization Practices and Sugarcane Yields Based on the Interviews

Each sugarcane farmer has their own schedule for fertilization. As shown in Table 2, fertilization can be conducted all year round, as long as fertilizers are available. Most farmers applied the first fertilization in January (23.7%) during the land preparation and sprouting period and the second fertilization in May (40.3%) during the tillering stage and the onset of the rainy season.

Chemical fertilizers were the major nutrient source and were applied by 90.82% of total surveyed farmers (see Supplementary File). Based on the survey, a variety of N:P:K ratios of chemical fertilizers were chosen, mainly N-rich types (see Table 2). Most farmers' choices relied upon the cost of fertilizers as the first priority, whereas adequate nutrient balance in soil and crop demand had not really been acknowledged in survey response. In this study, the amount of chemical fertilizers applied each year was 674.43 kg ha$^{-1}$ and delivered a cane yield of 73.1 ton per ha$^{-1}$ on average. Increases in fertilizer applied did not always increase the cane yield. As shown in Table 3, higher yields were found under low fertilizer applications <312.5 kg ha$^{-1}$ year$^{-1}$ for sandy soil, moderate applications of 312.5 to 625 kg ha$^{-1}$ for loamy soil, and the greatest application rates, 625 to 937.5 kg ha$^{-1}$, for clayey soil.

**Table 2.** Fertilization calendar and types of fertilizer applied in sugarcane plantations from the farmer survey (Number of observations, N = 207).

| | Jan | Feb | Mar | Apr | May | Jun | Jul | Aug | Sep | Oct | Nov | Dec |
|---|---|---|---|---|---|---|---|---|---|---|---|---|
| Growth stage | Land Preparation/Sprouting | | | Tillering | | Elongation | | | | Ripening | | |
| Percentage of farmers applied fertilizers | 23.7 | 12.7 | 18.2 | 17.7 | 40.3 | 13.8 | 10.5 | 13.3 | 12.1 | 7.2 | 2.2 | 1.1 |
| | Fertilizer application rate (kg ha$^{-1}$) | | | | | | | | | | | |
| Chemical fertilizer | 156–625 | 187.5–1250 | 62.5–937.5 | 104–625 | 104–937.5 | 156–469 | 156–625 | 156–625 | 156–1562.5 | 156–625 | 156–625 | 312.5–625 |
| N:P:K formula | 15:15:15, 16:8:8, 18:8:8, 21:0:0, 24:8:7, 25:7:7, 46:0:0 | 15:15:15, 16:20:0, 16:8:8, 21:0:0, 21:21:21, 30:0:0, 46:0:0 | 15:15:15, 16:20:0, 16:8:8, 21:0:0, 25:7:7, 30:10:10, 46:0:0 | 15:15:15, 16:8:8, 21:0:0, 46:0:0 | 12:20:0, 15:15:15, 16:16:16, 16:8:8, 21:0:0, 27:12:6, 30:0:0, 46:0:0 | 15:15:15, 16:16:16, 16:8:8, 21:0:0, 23:16:0, 30:0:0, 46:0:0 | 15:15:15, 16:20:0, 21:0:0, 46:0:0 | 15:15:15, 16:16:16, 16:8:8, 18:8:8, 21:0:0, 25:5:15, 30:0:0, 46:0:0 | 16:8:8, 21:0:0, 46:0:0 | 21:0:0, 21:21:21, 24:8:7, 46:0:0 | 21:0:0, 21:21:21, 24:8:7, 46:0:0 | 16:8:8, 46:0:0 |
| Manure (Chicken, Goat, Swine) | 312.5–3125 | 312.5–3125 | 312.5–46,870 | 312.5 | 312.5–3125 | Co-applied with chemical fertilizers | 312.5 | Co-applied with chemical fertilizers | Co-applied with chemical fertilizers | 312.5–3125 | NR | NR |
| Filter Cake | 310–156,000 | 120–156,000 | NR | NR | 250 | NR | NR | NR | NR | NR | NR | NR |
| Other organic fertilizer | 312.5 | 312.5 | NR | 312.5 | 156–625 | 625 | 156 | 625 | NR | NR | NR | NR |

Note: NR = Not reported.

**Table 3.** Median cane yield and cropping area observed for different soil textures and fertilizer amounts.

| Dominant Soil Texture | Fertilizer Amount, kg ha$^{-1}$ year$^{-1}$ | Median | | Number of Observations |
|---|---|---|---|---|
| | | Cane Yield, Ton ha$^{-1}$ | Crop Area Per Farmer Household, ha | |
| Sand | <312.5 | 81.3 | 625 | 5 |
| | 312.5–625 | 62.5 | 1250 | 9 |
| | 625–937.5 | 62.5 | 2500 | 3 |
| | >937.5 | 62.5 | 1875 | 4 |
| Loam | <312.5 | 64.4 | 1250 | 7 |
| | 312.5–625 | 75.0 | 1250 | 14 |
| | 625–937.5 | 62.5 | 1250 | 7 |
| | >937.5 | 62.5 | 1875 | 5 |
| Clay | <312.5 | 75.0 | 844 | 22 |
| | 312.5–625 | 75.0 | 938 | 25 |
| | 625–937.5 | 81.3 | 1875 | 10 |
| | >937.5 | 68.8 | 1250 | 17 |

As shown in Figure 1, sugarcane yield was geographically dependent, probably attributed to water availability and local soil properties. Higher yields were found in Bo Phloi and Tha Maka districts, whereas the yields in Mueang Kanchanaburi and Tha Muang districts were noticeably lower. Section 3.4.3. discusses water sources and irrigation in the study area, another likely explanatory factor. From Table 3, cane yields in clay soils were greatest when fertilizer applications were 312.5 kg ha$^{-1}$ year$^{-1}$ or higher, whereas sandy soil tended to provide the highest yield at low fertilizer levels (<312.5 kg ha$^{-1}$ year $^{-1}$).

Thailand imports most of their chemical fertilizers and the price can be volatile based on global supply and demand. In 2022, during the survey, the cost of chemical fertilizers doubled average rates due to the Russia–Ukraine war and the rising price of crude oil. As shown in Table 3, many farmers with small cropping areas cannot afford the chemical fertilizers or apply a smaller amount (<312.5 kg ha$^{-1}$ year $^{-1}$), while wealthier farmers with larger cropping areas were generally able to afford more fertilizers (625–937.5 kg ha$^{-1}$ year$^{-1}$) for sandy and clayey soil, and even higher application rates for loamy soil (>937.5 kg ha$^{-1}$ year$^{-1}$). Many farmers co-applied the main chemical fertilizers with other supplementary fertilizers (see Supplementary File), including manure (15.46%), organic fertilizers (4.35%), molasses (1.45%), and filter cake from sugar industries (11.59%). Manure and organic fertilizers were applied on the fields with no restriction on quality and quantity.

Molasses is a by-product of sugar industries and has been used as an agricultural additive to enhance biodegradation in organic fertilizers and soil. Another by-product of the sugarcane industry is filter cake, the cane pulp from the juice clarification process. The cake can be used for soil conditioning, resulting in better physical properties and increasing soil carbon for extended durations. Based on the interviews conducted, one application of ~100 ton ha$^{-1}$ during soil preparation can improve soil physical properties for three years. The filter cake can be acquired from the industries for free, but the farmers must bear their own transportation costs, resulting in unequal distribution of the filter cake to the farmers.

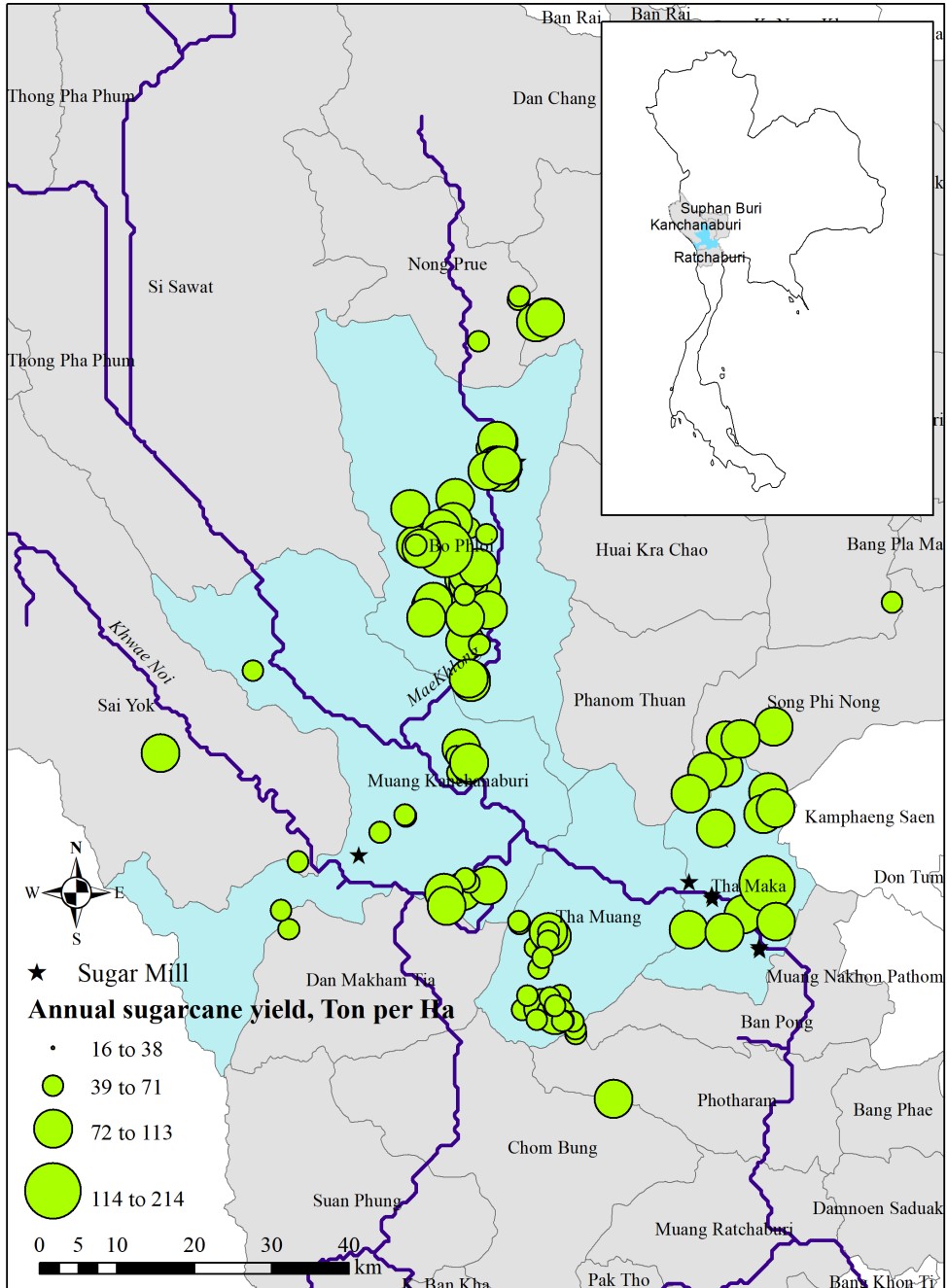

**Figure 1.** Spatial distribution of annual sugarcane yields acquired from the field survey conducted in May 2022.

### 3.2.2. Reviews on Using Waste from Sugar–Ethanol Industries as Fertilizer

### Treated Wastewater from Sugar Mills

Wastewater from sugar mills contains high total dissolved solid, TDS, (1480 mg $L^{-1}$), chemical oxygen demand, COD, (3682 mg $L^{-1}$), biochemical oxygen demand, BOD, (970 mg $L^{-1}$), and other nutrients [34]. With high organic matter and nutrients, sugar mill effluents serve as a good fertilizer [17]. Biological treatment of the wastewater is recommended for neutralizing wastewater's pH and reducing concentrations of sulfide, toxic metal ions, and other potentially harmful compounds [35]. Treated wastewater can be reused as biofertilizers [35]. Previous studies show that diluted, treated wastewater may

lead to increases in plant chlorophyll, carotenoids, total sugar, amino acids, and protein content and is not limiting for seed germination and growth [35].

Vinasse (Stillage, Spent Wash)

Vinasse can be considered as a liquid organic fertilizer with high K content. It contains 7–9% solids, of which 75% is organic and 25% is inorganic. The N content is mostly solid in form which acts as a slow-release fertilizer [15]. Considering its organic matter and nutrient content, the vinasse could be used as an alternative fertilizer and a source of organics. Though the nutrient composition of typical vinasse could be compatible with the mineral fertilizer, but it will be necessary to incorporate some nutrients from mineral fertilizer [36]. Moreover, direct applications of vinasse may cause detrimental short and long-term effects on the environment due to its high salinity, high organic load, and low P content [14].

The very high organic content in vinasse requires anaerobic treatment, anaerobic lagoon or anaerobic digestion with methane recovery. Methanation through anaerobic digestion can reduce COD by 58–80% [37], but the effluent still contains a high COD of about 15,000–46,000 mg $L^{-1}$. In addition, the effluent is darker due to melanoidin intensification [37]. The available technology for the anaerobic digestion of vinasse includes upflow sludge blanket, anaerobic fluidized bed reactor, anaerobic horizontal-flow immobilized biomass, and continuous stirred tank reactor [37,38]. Satyawali and Balakrishnan [27] suggest that to achieve maximal soil microbial activity, 10× dilution with freshwater for untreated vinasse and 2- to 16-fold for methanated vinasse is warranted. Fito et al. [8] recommended treatment via adsorption processes post-anaerobic digestion to provide additional removal of COD and color.

Filter Cake (Press Mud)

The filter cake is composed of a complex cellulose matrix and wax. Thus it requires the aid of microbes and a long incubation period for degradation [30]. The filter cake, however, contains sugar, which can easily decompose and enhance organic acid production in soil (Dotaniya et al., 2016). These microbially mediated organic acids can increase plant uptake of phosphorus in water-soluble forms and improve crop yield and quality [15].

Filter cake can be found with varying C/N ratios. High C/N ratio filter cake (>100) could promote nitrogen immobilization [15], whereas nitrogen mineralization may increase with an increasing incubation period [39]. Co-applications utilizing both the cake and inorganic fertilizer could remediate such limitations. Sharma et al. [40] suggested the integrated use of filter cake and urea (1:1) enhanced the cane yield in calcareous soil. Studies from various sites in India, China, Sudan, Swaziland, Egypt, Cuba, and Brazil agree that using filter cake in combination with chemical fertilizers increases crop yield and saves on fertilizer costs [31]. For those with a low C/N ratio of filter cake, mixing with sugarcane bagasse, which has a higher C/N ratio, might be beneficial for minimizing N losses [31].

Filter cake as a fertilizer can be composted with vinasse prior to field application [41]. Alternatively, filter cake can be mixed with cow manure and vermicomposted to increase nutrient availability [42]. Considering life cycle assessment, utilizing wastewater, ash, and filter cake from sugar mills as fertilizer could benefit the environment, especially by reducing natural resources consumption [6]. The waste to fertilizer conversion is estimated to reduce environmental impacts on human health by 1.2%, ecosystem quality by 0.28%, and minimize resource consumption by 31.2% [6]. Dotaniya et al. [15] recommended composting filter cake with vinasse in the ratio of 1:2.5 to obtain a biocompost enriched with ferrous sulfate ($FeSO_4$), zinc sulfate ($ZnSO_4$), and biofertilizer with N fixers and P solubilizers. This integrated use can mitigate the environmental problems from the direct application of vinasse on agricultural lands since the composting can provide higher soil phosphorus (P) content and lower salinity [14].

### 3.3. Applications of the By-Products and Wastes as Soil Conditioners

3.3.1. Vinasse (Stillage, Spent Wash)

Few farmers applied vinasse as a source of nutrients and water. As shown in Figure 2, the soils with wastewater applications were more acidic (<6.9) and tended to have a yellowish color (high hue in yellow-red shade). This characteristic is associated with the soil mineral composition, probably jarosite in acidic sulfate soil [43]. The acidic properties induced by wastewater application negatively impacts plant growth due to enhanced Aluminum (Al) and Mn toxicity and a decreasing nutrient capturing capacity of the soil grains. In addition, the wastewater does not provide benefits on water permeability in clayey soils. Long-term impacts associated with the vinasse applications were also found from the farmers' interviews, stating it resulted in poor soil physical properties. Nonetheless, it was recommended by the farmers that sandy soils are more tolerant, since wastewater application may lead to higher soil compaction [8]. Vinasse is also a source of K, Ca, and Mg, which can improve soil macro aggregation and increase sugarcane productivity [31]. In addition, introducing vinasse into dry sandy soils may increase soil enzyme activities, providing higher nutrient availability in soluble forms [30].

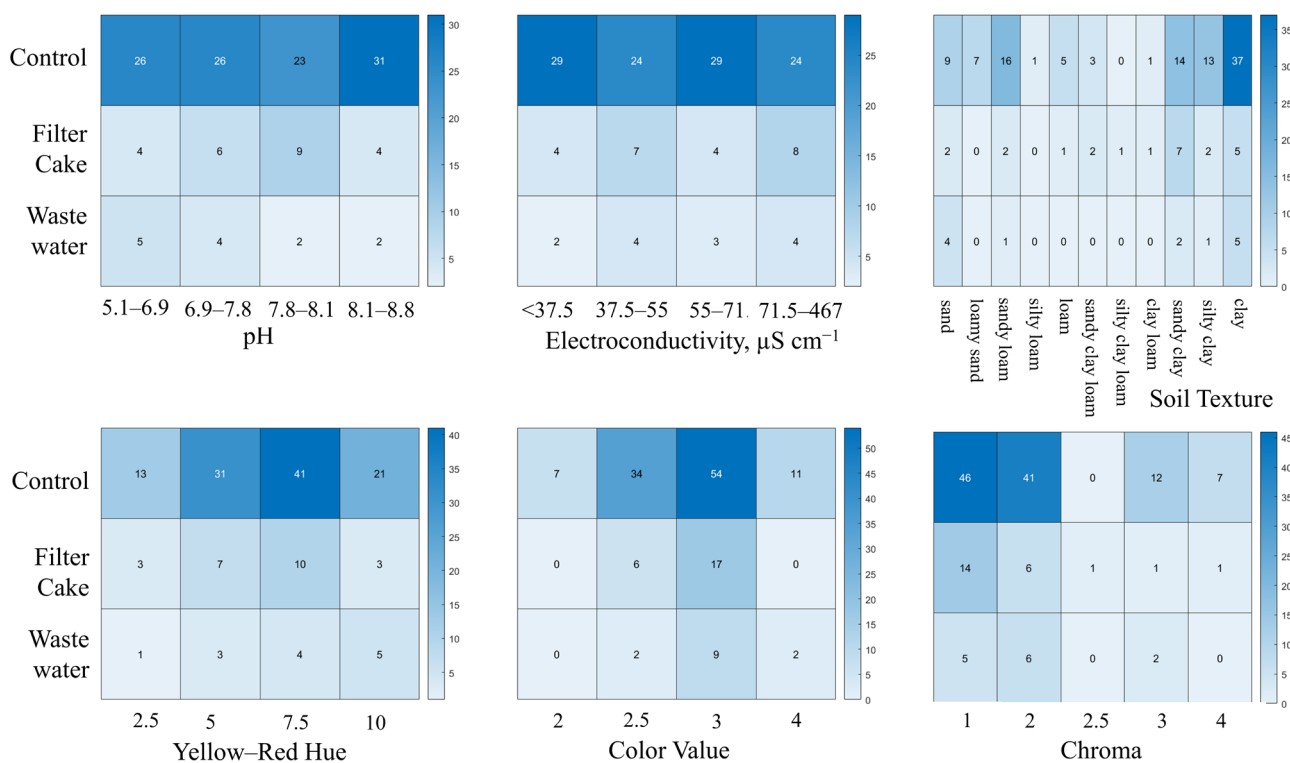

**Figure 2.** Number of observations (N) exhibiting different levels of pH, electrical conductivity, soil texture, Yellow-Red Hue, Color Value, and Chroma for sugarcane plantation soils with varying soil conditionings—filter cake application (N = 23), wastewater application (N = 13), and traditional practice or control (N = 106).

3.3.2. Filter Cake (Press Mud)

Filter cake application may also benefit soil physical and biological properties (Prado et al., 2013) [31]. Based on field observations of soil physiochemical properties in sugarcane plantations with different soil conditioning treatments (Figure 2), the soils with filter cake application generally exhibited higher water permeability, as implied from soil texture, and more mineral content, compared to control treatment soils. The filter cake-treated soil was more neutral (pH of 7.8 to 8.1) than that found in the control samples (pH > 8.1). However, the filter cake application was often tested in soils with sandy clay texture, whereas the control samples were clayey texture dominant. The soil with filter cake application was

also often found with low Chroma (=1) responding to more grayness, implying that the treated soils tended to have a greater underlying mineral content [44].

Filter cake contains fibrous material that has a high water-holding capacity. Nonetheless, fresh application is not good for soil and can be toxic to plants since it produces heat and ammonia during its rapid degradation. Thus biocomposting should be applied to extract wax and reduce heating and ammonia release from the cake before field applications [30].

Filter cake is produced through two main processes. Filter cake derived from the sulfitation process contains calcium sulfate ($CaSO_4$) and can be used for the amendment of alkaline soil or sodic soil [15]. Filter cake derived from the carbonation process contains calcium carbonate ($CaCO_3$) and can be used for acid soil amendment [15].

### 3.4. Feasibility of Using Wastewater for Ferti-Irrigation

3.4.1. Prospective of Farmers and Future Projections on Drought and Flooding

Drought significantly lowers cane yield within Thailand [3]. As shown in Figure 3, 44.7% of sugarcane farmers were highly impacted while 15.7% were extremely impacted by drought over the past 10 years. This suggests that more than half of the farmers could not adapt to drought conditions via their own resources and capabilities. Farmers who experienced high and extremely high levels of disruption were those who relied on water from rainfall, river runoff, ponds, and groundwater (Bo Phloi, Mueang Kanchanaburi, and Tha Muang sub-districts). Those who relied on irrigation water (Tha Maka sub-district) exhibited lesser drought impacts. For flooding, the impacts on the farmers' livelihood were significantly less with a substantial majority in the no-impact to moderate levels (~78%). The impacts were distributed across the study area without significant geographical dependence.

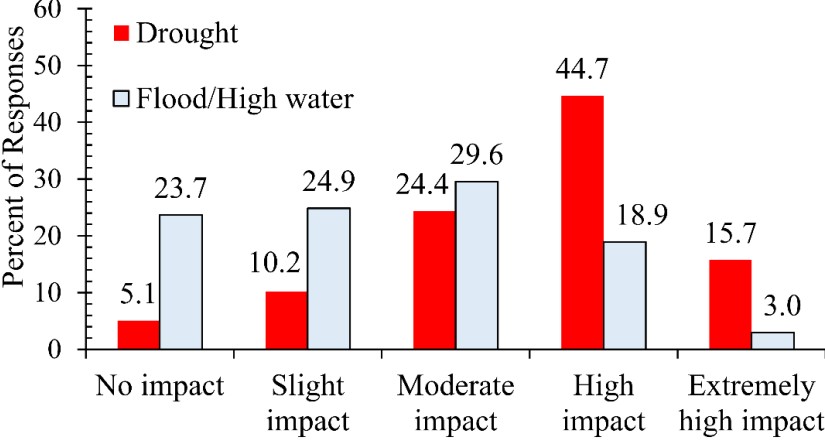

**Figure 3.** Percentage of sugarcane farmers experiencing impacts from drought (N = 197) and flood/high water (N = 169) at various degrees.

Increased temperatures could amplify the droughts in the dry season (November–April) in Thailand. A decline in annual precipitation across the country and further dryness was also reported in the work of Vongvisessomjai [45]. Typically, the irrigation water in this season relies upon stored water from the wet season. Therefore, dams are a crucial management tool for drought control [46]. Nonetheless, hydrological uncertainty makes dam operation challenging and it can become ineffective. It could even lead to major disasters in cases of dam operation failure. Prabnakorn et al. [47] assessed the archive climate data and concluded that increasing temperatures during the growing seasons will result in a higher possibility of rice yield losses.

3.4.2. Association between Cane Yield and Meteorological Parameters

Annual Cane Yield vs. Meteorological Factors

During sprouting and tillering in the dry season (February–May), lower rain and higher evaporation rates could increase yields (see the Pearson Correlation Coefficient, *r*, in Figure 4). Figure 4 also shows that the warming air temperatures at any cropping stage increased yield. This opposes what was found in the Lucknow and Uttar Pradesh, India, study by Samui et al. [48]. In this work, a maximum temperature (optimum of 26.8 °C), sunshine hours (optimum 9.2 h), and morning relative humidity of above 80% were beneficial for seed germination and enhanced cane yield. Later in the tillering stage, optimal minimum temperatures, afternoon humidity, and rainfall played a more significant role [48]. Cane yield increased with an increase in afternoon humidity. Rainfall of 96 mm was optimal for optimum yield while greater soil moisture stress reduced growth and yield [48]. A range of biological, chemical, and physical parameters may contribute to the different study outcomes observed with the different ecosystems studied. In our study in Kanchanaburi, Thailand, the dry seasons (2002–2020) had 68.3–75.2% monthly relative humidity, 7.73–144.8 mm monthly rainfall, 125.3–188.3 mm evaporation, 15.2–23.2 °C extreme minimum temperature, and 35.6–40.1 °C extreme maximum temperature (Figure 5). Thus, the data suggest that Kanchanaburi is comparatively hot and dry during the sprouting and tillering period. Nonetheless irrigated water is typically applied to cane fields to maintain the minimum requirements of soil water availability, particularly for triggering germination.

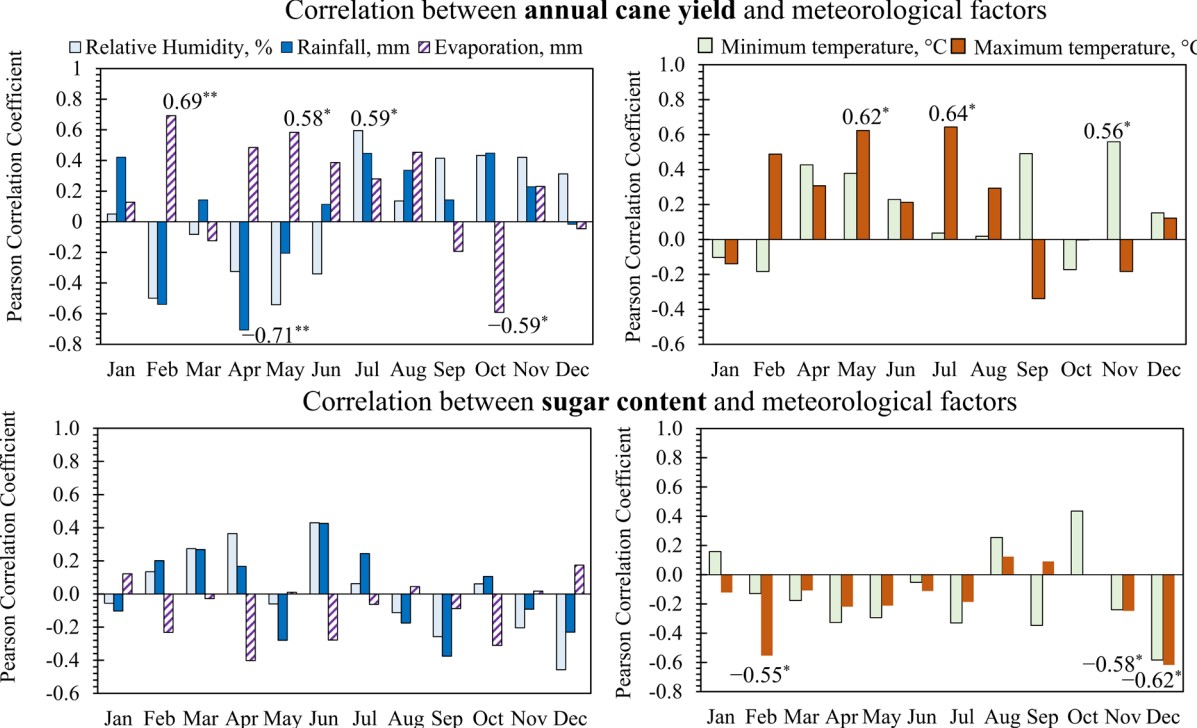

**Figure 4.** Pearson Correlation Coefficient, *r*, between annual cane yield and monthly meteorological parameters (**upper row**), and between cane sugar content and monthly meteorological parameters (**lower row**). The numbers with asterisks represent the *r* values exhibiting significant levels of * $p < 0.05$ and ** $p < 0.01$.

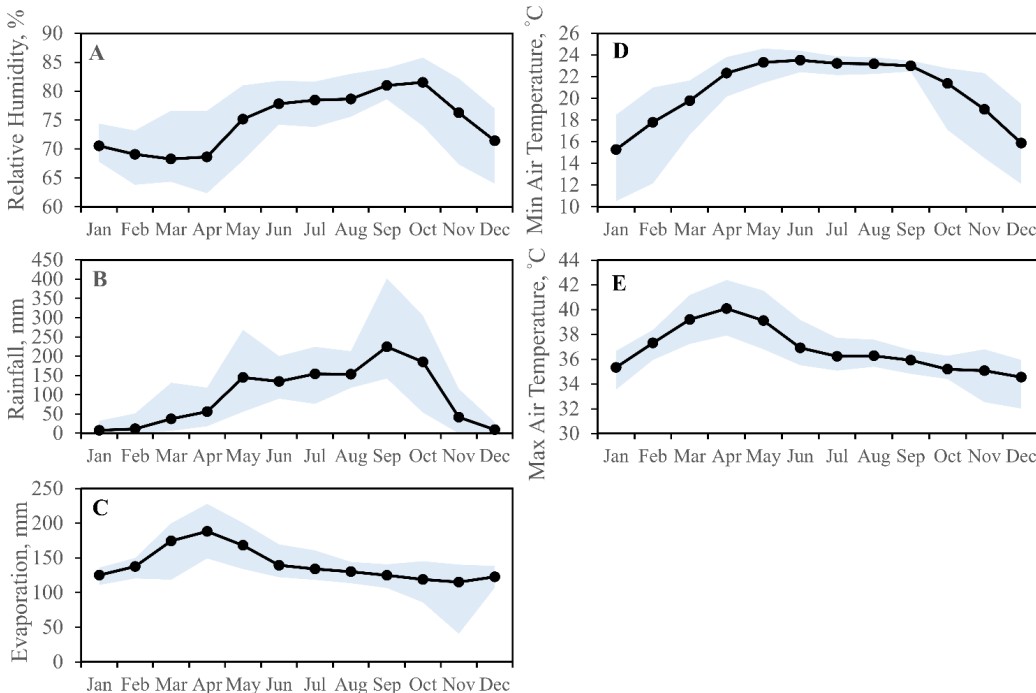

**Figure 5.** Monthly meteorological parameters (mean relative humidity—Panel (**A**), rainfall—Panel (**B**), mean evaporation—Panel (**C**), mean minimum temperature—Panel (**D**), and mean maximum temperature—Panel (**E**)) monitored from 2002–2020 at Ratchaburi station (424301 station code, 14.0114° N, 99.9678° E). The solid black line represents the mean values, and the blue area depicts the range from the minimum to the maximum values.

Atmospheric water, including rainfall and humidity, in the rainy season (July–October) positively corresponded with sugarcane yield (Figure 4). Significant positive correlation ($r = 0.59$, $p < 0.05$) was found between the humidity in July, when the cane was in the elongation stage, and final crop yield. Similar findings were reported by Samui et al. [48], who studied sugarcane agriculture in Uttar Pradesh, India. Their results showed that relative humidity and rainfall during elongation periods were positively associated with cane yield. Their conclusions, however, were strongly site dependent [48]. Scarpari and Beauclair [49] discussed further the best model for predicting sugarcane maturity. They suggested a model which incorporates precipitation, evapotranspiration, and soil available water capacity as primary variables and one which does not rely upon a single field factor.

Cane Sugar Content vs. Meteorological Factors

Lowering daily temperatures (both daily maximum and minimum) in December, the end of ripening period, significantly corresponded to higher cane sugar content ($p < 0.05$), as derived from Commercial Cane Sugar, CCS, (Figure 4; Pearson Correlation Coefficient, $r$, $= -0.58$ and $-0.62$ for maximum and minimum temperature, respectively). Similarly, Muñoz and Trujillo [50] monitored seasonal variations of the sucrose contents under sugarcane ripening in the Cauca river valley, Colombia and found that sucrose increased when minimum air temperature was lower than 18.5 °C. Scarpari and Beauclair [49], studying sugarcane in São Paulo, Brazil, found sucrose increased in stems with lower nighttime temperatures and reduced plant growth. Since temperature limits photosynthetic and respiration rates in sugarcane, lowering temperatures suppress respiration and photosynthesis, resulting in accumulation of carbohydrate and sucrose content [51]. In addition, Muñoz and Trujillo [50] found that low precipitation and a high vapor pressure deficit (high plant transpiration rate) leads to increased sucrose content. In our findings, while there was a negative correlation between precipitation/relative humidity and sugar content, the correlation was insignificant ($p > 0.05$).

### 3.4.3. Water Sources for Sugarcane Cultivation Based on the Farmer Interviews

To mitigate the effects of insufficient rainfall and runoff, farmers primarily use pumped groundwater (63.77%) or surface water ponds (14.49%). As shown in Table 4, the majority of farmers relied on groundwater for sugarcane cultivation, especially in Bo Phloi and Tha Muang Districts (Figure 6). In Muang Kanchanaburi District, groundwater was inaccessible, therefore the farmers relied on rainfall, surface runoff, and surface water ponds. Rainfall and surface runoff are predicted to become highly uncertain under climate change, and deficits are likely to increase in drought years, resulting in lower average yield for this area (see Figure 1). Irrigated water can be accessed for 13.5% of farmers surveyed, particularly in the region downstream of Mae Khlong River—Tha Maka District (see Figure 6), leading to greater cane yields (see Figure 1). Despite the farmers' perception that alternatives to irrigation water are needed, treated wastewater (4.35%) was only welcomed in plantations nearby the sugar mill in Bo Phloi District (see Figure 6). Many farmers have negative perceptions of the sugar mill wastewater on health, agriculture, and the ecosystem, as shown in Figure 7.

**Table 4.** Percentage of water sources for sugarcane cultivation based on the farmer interviews.

| Major Water Sources for Sugarcane Cultivation | Percentage (N = 207) |
|---|---|
| Rainfall | 38.65 |
| Groundwater | 63.77 |
| Surface Runoff | 14.98 |
| Industrial Wastewater | 4.35 |
| Irrigated Water | 13.53 |
| Pond | 14.49 |

Most of the sugar mills in Thailand are not legally allowed to discharge treated wastewater directly into natural waterways. The wastewater is currently reused in the sugar mill for cleaning and gardening purposes or discharged to nearby shrublands. In 2020, regional drought was sufficiently extensive and agriculturally impacting that, in response, the Ministry of Industry, Thailand issued the Notification of the Ministry of Industry on "Criteria, methods and conditions for approving the use of industrial wastewater for temporary use in agricultural areas during drought year 2020." The announcement enforced new Ministerial Regulations No. 26 (B.E. 2563), issued under the Industry Act B.E. 2535. These regulations determined the feasibility of using dilution methods for discharging industrial wastewater under certain criteria. This solution is intended to mitigate drought problems in agricultural sectors; however short-term and long-term agricultural, ecological, and social impacts are currently unclear.

In the case of molasses-based ethanol distilleries, water footprint analysis in Kanchanaburi Province, Thailand shows that 849.7 m$^3$ water ton$^{-1}$ was taken from rain and surface water, 209.6 m$^3$ ton$^{-1}$ from irrigating water, and 45 m$^3$ ton$^{-1}$ as wastewater [52]. Wastewater, however, was stored in ponds or reused in the industry, and not discharged into the environment [52].

### 3.4.4. Perception of the Farmers on the Treated Wastewater Quality

About 33.5% of the farmers interviewed were not confident about using the treated wastewater as irrigation water, whereas 25.7% were moderately confident with some concerns, and 18.4% were highly confident. Perceptions of farmers regarding treated wastewater quality were diverse. Some farmers responded that treated wastewater provided no benefits to cultivation due to limited nutrient availability and high transportation costs. Many farmers were unsure about treated wastewater characteristics but have experienced rancid and turbid brownish water from the sugar mill (see Figure 7A). The rancid smell can adversely affect the respiratory system, impacting farmworker health (see Figure 7B). Many farmers agreed that wastewater would increase sugarcane yield and quality but would likely degrade soil quality over time (see Figure 7C). Furthermore, concerns were raised in the interview that discharging the water to

natural channels would degrade surface water quality, resulting in a dead zone with impacts on local ecology and biodiversity (see Figure 7D).

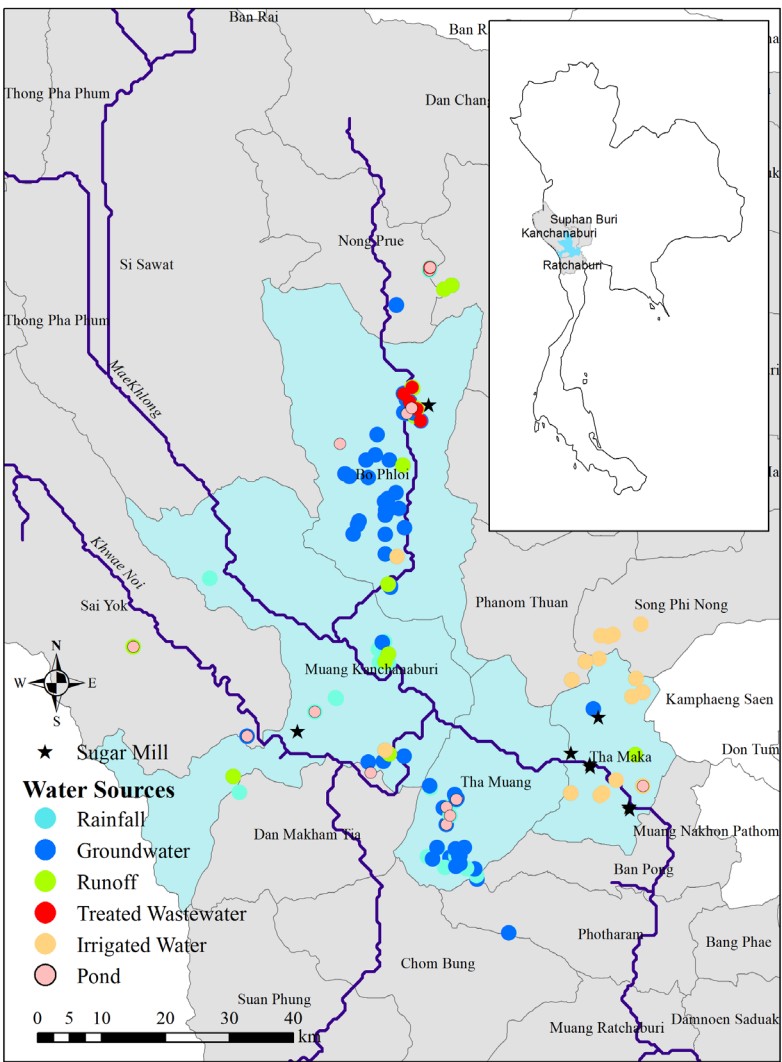

**Figure 6.** Spatial distribution of types of water sources acquired from the farmers' interviews. The blue area represents the four studied districts. The star symbol represents the location of sugar mills in the studied area.

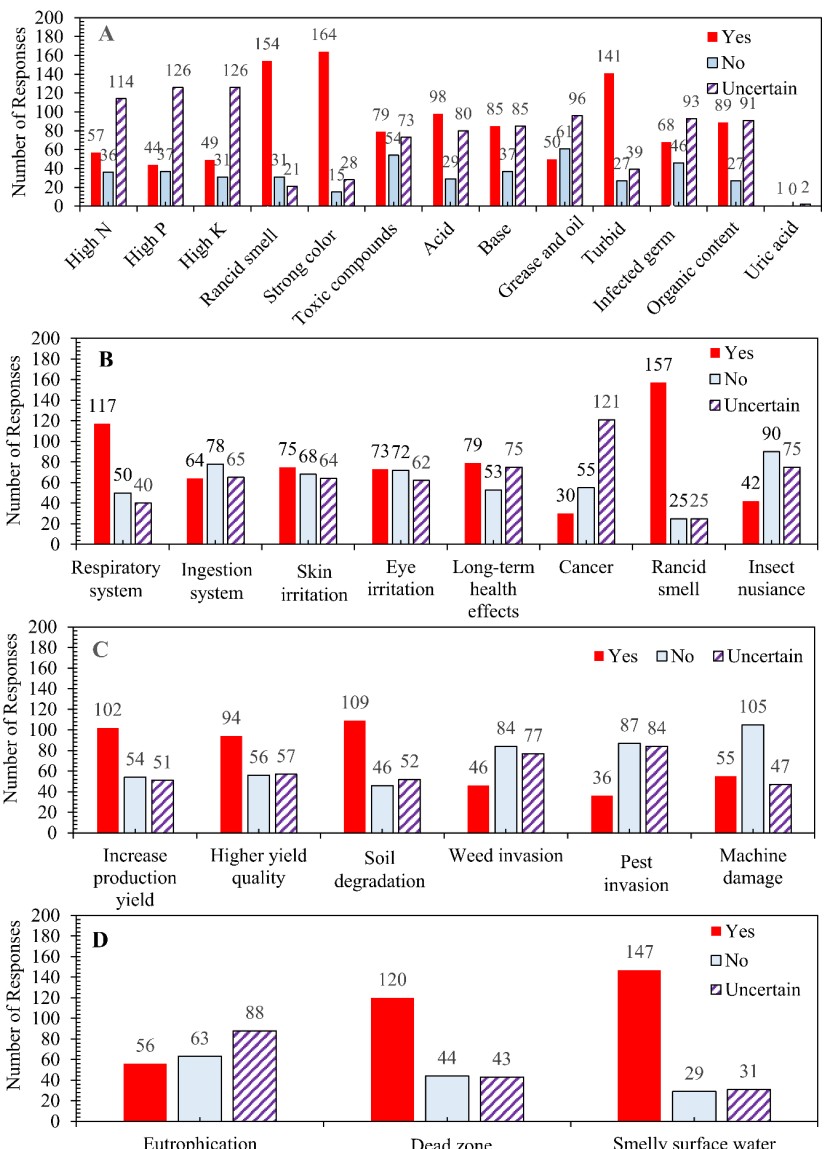

**Figure 7.** Farmers' perceptions of the treated wastewater quality from local sugar mills and the impacts. Panel (**A**) represents perception on properties of treated wastewater, Panel (**B**) represents perception on health impacts, Panel (**C**) represents perception on agricultural impacts, and Panel (**D**) represents perception on ecosystem impacts.

### 3.4.5. Using Wastewater as Ferti-Irrigation Water Based on Literature Reviews

After biological treatment, the sugar mills' wastewater will have an improved appearance, smell and, quality, with neutral pH, low organic contents, low sulfide, low salt, and low toxic metals [35]. Singh et al. [35] recommend that this type of treated water can be used for ferti-irrigation, which provides improved crop yield. Saranraj and Stella [17] support irrigation using effluent from sugar mills, focusing on effluent provision of nutrients for plant growth, improvement of soil structure, and increase in crop yield.

Vinasse from ethanol distilleries should not be directly applied to fields as irrigation water. It requires anaerobic treatment to reduce salt and organic loading, especially melanoidin which results in a dark color [14,37]. Diluting vinasse is recommended. Kumar and Chopra [53] studied effects of vinasse concentrations on plant (*Trigonella foenu-graecu*) growth. Their findings show that growth parameters and yield increased with vinasse concentration (from 5% to 50%) but decreased at higher concentrations (75% to 100%). Heavy metal accumulation also increased for both soil (Cd > Cr > Pb > Zn > Cu > Fe) and

plants (Pb > Cr > Cd > Cu > Zn > Fe) after irrigation with vinasse [53]. Thus, optimal dilution of vinasse should be further investigated to minimize the accumulation of toxic substances in soil and biomass and maximize plant growth.

*3.5. Challenges and Practical Implications on Waste Utilization for Agricultural Production*

This section delineates key challenges for future research and implementation regarding waste utilizations for sustainable agriculture. Additionally, it provides practical implications for future practices and studies, which could be beneficial for policymakers, researchers, and farmers to consider in advancing the BCG initiative in the sugar–ethanol industry in Thailand.

### 3.5.1. Financial Feasibility

In Thailand, the waste or by-products from the sugar mills (bagasse and molasses), are mostly used for energy production, which has an immediate economic advantage and is effectively running within the proposed loop of the BCG. Utilizing filter cake and wastewater (both treated wastewater and vinasse) as irrigation water are considered in this study since they can be potentially returned to the field and create benefits for farmers. Transporting large amounts of wastewater and filter cake from industries to plantations is, however, the greatest and most identified challenge by the farmers. This is the most critical barrier for drought mitigation through waste utilization.

***Treated wastewater***: Based on farmer interviews, the majority of farmers preferred the transport of treated wastewater via irrigation channels (95%), direct pipeline (41%), or surface runoff (9%). Drought is a key factor resulting in national cane yield decline and farmers in Kanchanaburi experience very high impacts associated with drought, with many farming households taking a long time to recover economically. Using industrial wastewater, which is properly treated and transported through the irrigation channels, could more or less alleviate drought impacts in agricultural lands near local industries, but requires legal permission. In more distant crop fields, many farmers did not see any advantages to using the industrially treated wastewater, unless the water contained some nutritious substances required by the plants. A main concern of farmers was that the cost of transportation might overwhelm the enhanced profit from greater yield, leading to an overall economic loss. Irrigation water containing, or supplemented with, nutrients for plant growth, or "ferti-irrigation," is deemed to be more economically sound. Thus, incorporating treated wastewater into the field irrigation as a drought mitigation measure must consider approaches that pre-treat the wastewater to reduce/remove toxic components and maintain optimum nutrient content for plant growth.

***Vinasse and filter cake***: Both the published literature and local farmer survey agree that vinasse and filter cake can be applied to sugarcane plantations to reduce the use of chemical fertilizers and could benefit cane production. To reduce the cost of chemical fertilizers, some farmers have already begun to apply vinasse directly in the field. The vinasse has typically been delivered to the farmers' fields on demand and the farmers must bear their own transportation costs. Nonetheless, due to legitimate concerns, industries may set certain conditions or prioritize certain groups (such as organic farming or quota farmers) to ensure no problems are associated with improper waste management. Furthermore, wastes or by-products need to be pre-treated before use to maximize their benefits and minimize impacts on agroecosystem health.

### 3.5.2. Laws and Regulations

In most cases, discharging sugar mill-generated and -treated wastewater into natural waterways is not permitted by law in Thailand. However, such discharge may be periodically allowed under drought crisis management by the Ministry of Industry. Wastewater would be directly delivered to farmers' fields by request, typically using available water tracks. There are many governmental agencies involved in delivering this legislation, which may act as another barrier to transporting treated wastewater to the field by effective means.

The involved agencies include the Ministries of Industry, Natural Resources and Environment, Agriculture and Cooperatives, and Public Health. Thus, all relevant agencies should work together to enhance public adaptability to climate change, sharing infrastructure, financially supporting common goals, and collaborating on policy development.

Utilizing waste and by-products from sugar mills and molasses-based ethanol distilleries will not be implemented without regulatory support. Waste solutions are currently delayed, in part due to ambiguous and overlapping definitions of wastes and by-products. Industrial waste, currently identified as hazardous waste, must be appropriately treated and managed by certified agencies by the Ministry of Industry. Thus, industrial waste utilization for agricultural production is not feasible under the current definition. Some industries, therefore, utilize their own wastes within their own business networks, including energy and ethanol production. The term "by-products," however, is used as a substitute for "industrial waste" to avoid the difficulties caused by legal violations. Different perspectives from various stakeholders on wastes and by-products challenge the improvement of the role of the industrial sector on the BCG through waste utilization. Once utilization of wastes and by-products becomes more profitable, registrations and supply chains must be prepared for distributing benefits to allow maximum sustainability in driving the BCG.

### 3.5.3. Farmer Perception and Practices

Based on the survey results concerning utilizing wastewater in Kanchanaburi, Thailand, local farmers perceive significant adverse health impacts (respiratory system and unpleasant odor), agricultural impacts (soil degradation), and water quality degradation. Though farmers showed higher productivity associated with wastewater utilization, the majority of the farmers avoided applying the wastewater to their fields due to concerns over long-term soil degradation.

Most farmers were uncertain about the properties of the treated wastewater and merely identified it as vinasse or "wastewater" as a whole. Contrary to field-based results, farmer understanding of field fertilization is driven by the perception that the more fertilizer applied, the more yield is obtained. Many farmers used no specific type of chemical fertilizers for cropping and were therefore less likely to create ideal soil nutrient conditions in their fields. Based on the survey, yield was not strongly associated with the amount of chemical fertilizer applied, but was more geographically dependent, probably attributed to water availability and soil properties.

To promote BCG implementation, knowledge on waste utilization, proper fertilization, and cost effectiveness analysis should be disseminated to all farmers. Learning centers on waste utilization should be established and supported by the government under a collaborative effort with industrial partners.

### 3.5.4. Sound Agricultural Practices

Once biomass cane has been harvested off the field, nutrients are depleted, and additional sources of nutrients, mainly chemical fertilizer, must be provided for the next cultivation. Thus, returning wastes or by-products from sugar mills or distilleries (in the form of filter cake and wastewater) could be partly fulfilling carbon and nutrient cycles in the agroecosystem. Here are suggestions on waste utilization to maximize the agricultural benefits for the farmers, as follows:

- Treated wastewater from the sugar mills can be released into the irrigated waterways and delivered to the nearby fields. The water quality must be continually monitored by relevant parties: the Department of Industry, Pollution Control Development, and the Royal Thai Irrigation Department.
- Filter cake is typically applied to improve soil physical properties and also reduce fertilizer use. Filter cake must be biodegraded and fermented, probably with vinasse, before being applied to the field, which can increase bioavailability, increasing nutrient availability, and avoid negative impacts to the soil and plant [15,30]. Since it has

comparatively low available N, the treated filter cake should be applied together with urea fertilizer, cow manure, or bagasse [31,40,42].

- Vinasse is K- and N-rich. Thus, it has been recommended as an alternative source of plant nutrients. Nonetheless, it should not be used without being pre-treated, probably by anaerobic fermentation to reduce organic loading. Dilution is strictly recommended to avoid negative impacts on soil quality and to enhance soil microbial activity [27,53]. Furthermore, its application to sandy soils could be of the greatest benefit as suggested by the farmer survey.

### 3.5.5. Environmental Health

Long-term applications of vinasse can potentially affect soil–aquatic ecosystems. Excessive applications of vinasse can result in soil aggregation, resulting in soil alternations, including improved aggregation and increased soil water infiltration. This could promote subsurface water contamination [31]. Consistently, da Silva et al. [54] reported organic contaminants leached into the groundwater were associated with the repeated application of vinasse. Fuess and Garcia [16] also reported adverse effects of vinasse ferti-irrigation, including soil and groundwater acidification and soil salinization. In our survey, many local farmers perceived that potential surface water degradation and pungent smells were attributed to the direct discharging of wastewater or vinasse on the crop fields. The proper treatment of vinasse could reduce toxicity by 98.6%, which is sufficient for transforming the vinasse to an organo-mineral fertilizer [36]. Yin et al. [55] also reported low ecological risk of the toxic metals in the vinasse and negligible accumulation of heavy metals in agricultural soil via long-term irrigation. Thus, the wastewater, especially vinasse, should be treated and carefully utilized in proper time, duration, and magnitude. Assessing long-term ecosystem impacts, including soil, biomass, and groundwater, should be further implemented to affirm a healthy waste utilization environment.

Vinasse emitted greenhouse gases during storage and during transportation by open channels [56]. The methane ($CH_4$) emissions ranged from 394 to 1092 mg m$^{-2}$ h$^{-1}$ and 1.36 kg $CO_2$ equivalent m$^{-3}$ of transported vinasse [57,58]. Anaerobic digestion of the vinasse is recommended to capture the biogases, mainly $CH_4$, for energy recovery, as well as to reduce the organic loading before being applied to the sugarcane fields [8,27,28,59].

It is noted that agricultural open burning, resulting in air pollution problems, had been an issue in many parts of Thailand. Farmers usually burnt sugarcane plantations to ease harvesting. The practice is illegal, but law enforcement alone cannot effectively solve the problem. To mitigate adverse effects from biomass-burning smoke, the Thai government has employed both strict law enforcement and a positive incentive policy, subsidizing those who cut fresh cane. According to this legal employment, the proportion of burnt cane significantly declined from 65% as normal range to 30% in the 2020/2021 production year. The evidence can affirm that effective policy can solve the problem. However, there has been concern that without governmental subsidy, the burning and air pollution will return.

### 3.5.6. Value-Added Products

Vinasse and filter cakes, apart from their direct applications in agriculture, possess potential for producing value-added bioproducts. This includes furfural, which can be further converted into various chemicals used in petrochemical, plastic, pharmaceutical, and agrochemical industries [60] (Santos et al., 2020). Several studies have revealed the possibility of using alcohol vinasse in the production of biofuels, biomedicine, food, cosmetics, and packaging [61] (Montiel-Rosales et al., 2022). The wax content in filter cakes can meet industrial requirements for use in the food, pharmaceutical, chemical, cosmetic, cleaning, and polishing industries [60] (Santos et al., 2020). If these technologies can be scaled up for commercial and industrial uses, bioproducts may offer more economically viable alternatives to agricultural applications, thereby promoting the BCG economy. Other by-products from sugar–ethanol industries, namely bagasse and molasse, which are not considered as industrial wastes in this study, could also be used to produce value-added

bioproducts through biorefinery processes. These processes involve fermentation to produce alcohols and organic acids like adipic acid [62], digestion to generate biogas and biofertilizers, and an enzymatic process to release sugars (monosaccharides, hexoses, pentoses, glucose, xylose, etc.) present in lignocellulosic biomass [60]. Previous studies have shown positive progress in using cane residues for biorefinery processes, resulting in a wide range of value-added products. However, there are concerns that current technologies may be expensive, slow, and environmentally unfriendly [60,61]. Therefore, intensive research into advanced technologies for bio-based product manufacturing and financial feasibility is essential.

### 3.5.7. Comprehensive Sustainability Assessment

A sustainability assessment of waste utilization practices should be exercised to quantify their environmental footprint, analyze emissions, energy use, and resource consumption throughout the entire process, considering various input scenarios. The assessment can guide the sugar–ethanol industry towards sustainable practices. It provides the necessary information and insignts to make informed decisions, optimize processes, and align activities with broader sustainability objectives. Through the assessment, industries can work towards both environmentally responsible and socially beneficial approache to waste utilization. The assessment can be accomplished through integrated exergy-based or life cycle assessment approaches [63,64].

### 3.6. Research Limitations

While the study offers valuable insights into waste utilization in the sugar–ethanol industry and its potential benefits for sustainable agriculture, it is important to acknowledge certain limitations. The findings are based on data and observations in a specific region and may not be directly applicable to other regions with different environmental, socioeconomic, and agricultural conditions. This study incorporates feedback from farmers but it may not capture the perspectives of all relevant stakeholders, such as policymakers, industries, environmental agencies, and local communities. A more comprehensive understanding could be achieved through broader stakeholder engagement. Additionally, this study may not fully address the socio-economic aspects of waste utilization and its long-term impacts, which could influence the acceptance and adoption of waste utilization practices by local communities and stakeholders.

### 4. Conclusions and Prospects

Thai sugar is a significant contributor to the world sugar market and it provides economic security to the nation. National BCG initiatives promote the diversification of the by-products and wastes from sugar mills to food production (such as irrigated water, fertilizer, and soil conditioner). This could contribute to augmenting farmers' livelihoods and environmental health. This study aimed to assess the current situation and feasibility of applying by-products from sugar mills and distilleries back into the sugar plantations for irrigation, fertilization, and soil conditioning. This study is based on reviewing literature worldwide, together with preliminary surveying and analyzing samples taken from the fields in Kanchanaburi, Thailand.

Appropriate fertilization with treated wastewater, vinasse, and filter cake can enhance plant growth when co-applied with chemical fertilizers, as they contain some nutrients and mineral sources. The vinasse proves especially advantageous for sandy soils as it enhances enzyme activities, soil aggregation, and nutrient availability. Conversely, prolonged application in clayey soils may lead to soil compaction. Bio-composted filter cake generally improved soil water permeability and increased mineral contents, typically when applied to resolve highly compacted soils.

Approximately 60.4% of farmers face considerable vulnerability to drought, resulting in reduced cane yield (associated with low atmospheric water levels during elongation stage) and reduced sugar content (associated with elevated temperatures during ripening

stage). Given that the treated wastewater aligns with national irrigation water quality standards, using ferti-irrigation with the treated wastewater could be an option for mitigating the drought impacts. However, roughly 33.5% of farmers were not confident about using treated wastewater for irrigation due to some expressing concerns about limited nutrient availability, high transportation costs, and negative impacts on health, agriculture, and the ecosystem.

While the utilization of waste could augment sustainable agriculture in Thailand, several avenues for future implementation and research must be explored to achieve the sustainable goal. In terms of financial feasibility, transportation cost is the main issue. Most wastes and by-products are transported by road, and the expenses must be borne by the farmers. It is recommended that a feasibility study be undertaken to ascertain the viability of employing irrigation canals as a distribution method for treated wastewater, particularly in mitigating drought in farmlands.

Moreover, researchers should investigate the conversion of waste materials into value-added bioproducts, which present more financial benefits. Within the regulatory aspect, ambiguous definitions of by-products and wastes create additional obstacles to the implementation of BCG. Those involved in waste utilization must also comply with national laws and work with many responsible agencies, particularly those involved in industrial waste management. Policymakers can use the findings to formulate policies that support and incentivize waste utilization practices. At the local government level, facilitation of industrial engagement and partnerships with educational institutions are crucial for disseminating knowledge on waste utilizations to farmers. The majority of farmers lack sufficient understanding regarding proper fertilization and treatment of the wastes before use. Many farmers also experienced negative impacts from the waste utilizations. Additionally, long-term monitoring of environmental impacts and sustainability assessments on the waste utilization should be implemented to ensure sustainability.

**Supplementary Materials:** The following supporting information can be downloaded at: https://www.mdpi.com/article/10.3390/su152014711/s1, Figure S1: Percentage of sugarcane farmers applying various types of fertilizers.

**Author Contributions:** A.B.: conceptualization, methodology, collecting data, formal analysis, writing, and funding acquisition; N.K.: co-writing and field sampling; J.K.: field surveying, field sampling, and project administration; R.F.: supervision; B.S.: sample analysis; J.P.J.C.: supervision; K.R.R.: supervision, writing, administration, and funding acquisition. All authors have read and agreed to the published version of the manuscript.

**Funding:** This research was financially supported by the British Council Research Environment Links in joint venture with the Program Management Unit for Human Resources and Institutional Development, Research, and Innovation (PMU-B), Thailand year 2021/22.

**Institutional Review Board Statement:** The study was conducted in accordance with the Declaration of Helsinki, and approved by the Institutional Review Board (or Ethics Committee) of Mahidol University (protocol code MU-CIRB 2022/038.2302 since 28 April 2022).

**Informed Consent Statement:** Informed consent was obtained from all subjects involved in the study.

**Data Availability Statement:** All data generated or analyzed during this study are included in the manuscript.

**Acknowledgments:** This research received financial support from the British Council in joint venture with the Program Management Unit for Human Resources and Institutional Development, Research and Innovation (PMU-B), Thailand, under the Research Environmental Link year 2021/22. The Research and Academic Service, Mahidol University Kanchanaburi Campus provided the support for laboratory services. Special thanks are given to Peerata Khunoth, the laboratory technician, for her service. We appreciate the kind assistance from the Cane Planters Association Zone 7 (Naratip Anuntasuk), Thamaka Sugar Industry (Karunchai Taksin), and KSL (Bo Phloi) Sugar Industry.

**Conflicts of Interest:** The authors declare no conflict of interest.

## Abbreviations

List of abbreviations, symbols, and nomenclatures used in the manuscript.

| Abbreviation | Definition |
|---|---|
| BCG | Bio-Circular-Green Economy |
| GDP | Gross Domestic Product |
| CCS | Commercial Cane Sugar |
| pH | Power of Hydrogen Ion |
| TDS | Total Dissolved Solid |
| DO | Dissolved Oxygen |
| EC | Electroconductivity |
| TN | Total Nitrogen |
| TOC | Total Organic Carbon |
| IC | Inorganic Carbon |
| BOD | Biochemical Oxygen Demand |
| COD | Chemical Oxygen Demand |
| TKN | Total Kjeldahl Nitrogen |
| C | Carbon |
| N | Nitrogen |
| Fe | Iron |
| Mn | Manganese |
| Zn | Zinc |
| Cu | Copper |
| Cr | Chromium |
| Pb | Lead |
| Cd | Cadmium |
| Hg | Mercury |
| As | Arsenic |
| Ca | Calcium |
| Mg | Magnesium |
| K | Potassium |
| P | Phosphorus |
| Si | Silica |
| Al | Aluminum |
| $FeSO_4$ | Ferrous Sulfate |
| $ZnSO_4$ | Zinc Sulfate |
| $CaCO_3$ | Calcium Carbonate |
| PCU | Platinum Cobalt Unit |
| NTU | Nephelometric Turbidity Unit |
| *r* | Pearson correlation coefficient |
| *p* | *p*-value |
| B.E. | Buddhist Era |

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
