# Peer review of "Balancing Agriculture and Industry through Waste Utilization for Sugarcane Sustainability"

_sustainability, doi:10.3390/su152014711_

Round 1

Reviewer 1 Report

This review talks about the implementation of the Bio-Circular Green Economy on the sugarcane industry in Thailand. It's overall a well-written review.  

The English needs improvement because there are some typos. I have uploaded the PDF with some comments and highlights.

Author Response

Responses to the reviewers on the manuscript of “The challenge of Waste-to-Foods in the Sugar-Distillery Industry in Driving Bio-Circular Green Economy.”

Reveiwer1

We are thankful for your comments. All the highlights are considered and resolved. The improvements are shown in red and listed here.

  • About 3,885 G Wh electricity or about 2% of total electricity production in Thailand was derived from bagasse in 2016 (Chunhawong et al., 2018; IAEA, 2023).
  • Change from “Due to changes in the climate, amounts of rainfall and surface runoff are highly uncertain…” to “Rainfall and surface runoff are predicted to become highly uncertain under climate change,….”
  • Correct writing from “yeild” to “yield”, from “pipepline” to “pipeline”, from “advanatges” to “advantages”, from “ discharing” to “discharging”

Reviewer 2 Report

The manuscript is well-written and contains a detailed study. Overall, I recommend the manuscript for publication. Following are a few minor suggestions that can further improve the manuscript.

1.      Authors can also check for toxic compounds in the wastewater and Vinasse so that this study's phytoremediation potential can also be analyzed.

2.      Fig 7. Crosscheck for the axis title.

Minor editing of English language required

Author Response

Responses to the reviewers on the manuscript of “The challenge of Waste-to-Foods in the Sugar-Distillery Industry in Driving Bio-Circular Green Economy.”

Reviewer 2

  1. Authors can also check for toxic compounds in the wastewater and Vinasse so that this study's phytoremediation potential can also be analyzed.

Response        Thank you for your suggestions. We did analyze the metal contents in the vinasse, but not the wastewater, since the health and ecological risks of the vinasse are more controversial. Previously, We hesitated to show the metals analysis results since the analysis is not with multiple replicas. As you suggested, we delivered the results, along with the other literature findings. In such a way, the findings should be more reliable.

Line 235-245:

“However, the magnitudes could exhibit signigicant variation. Our findings for the post-methanated vinasse indicate that the Fe and Mn could be considerable (19.05 and 8.91 mg kg-1, respectively), but Zn (0.61 mg kg-1) and Cu (undetectable) were notably lower compared to the vinasses studied by Rajkishore and Vignesh (2012), as well as lower than the typically found in organic fertilizers, such as manures and mushroom substrate, reported by Kuziemska et al. (2023).

The levels of toxic metals in the post-methanated vinasse were considerably low: 0.04 mg Cr kg-1, 0.08 mg Pb kg-1, 0.27 mg As kg-1‑, <0.001 mg Hg kg-1, and non-detectable Cd. These levels are well below the threshold levels outlined in the national guideline for liquid biofertilizer (5 mg Cr kg-1, 3 mg Pb kg-1, 3 mg As kg-1, 0.5 mg Hg kg-1, and 2 mg Cd kg-1), as established by Land Development Department (2013).”

  1. Fig 7. Crosscheck for the axis title.

Response        Thank you for your suggestion. The Y-Axis title is now provided.

Reviewer 3 Report

Please find attached file 

The manuscript should have extansive English editing to have a clear presentation 

Author Response

Responses to the reviewers on the manuscript of “The challenge of Waste-to-Foods in the Sugar-Distillery Industry in Driving Bio-Circular Green Economy.”

Reviewer 3

  1. Please clearly describe the problems for using byproduct into agriculture production that should be solved in the experiment in the introduction.

Response        In the introduction, we present the findings from multiple literatures on how sugar mills could improve sugarcane value chains to contribute to the BCG and sustainability goals. We add two sentences providing potential adverse impacts associated with waste utilization.

Line 94-100

“While the byproducts from the sugar-ethanol industries can be used for agriculture, there are potential negative impacts associated with its utilization. The impacts include soil salinity (Murugaragavan and Mahimairaja, 2009; Murillo et al., 1993), disturbance in soil pH (Dotaniya et al., 2016), groundwater contamination (Fuess and Garcia, 2014), the accumulation of heavy metals in soils (Saranraj and Stella, 2014), the inhibition of growth in terrestrial plants and aquatic organisms (Sousa et al., 2019), alterations in soil microorganisms (Moran-Salazar et al., 2016), and problems associated with regulatory compliance.”

  1. The manuscript only describes to solve the problem using fermentation, but the technology for the fermentation is not explained. The author suggested that utilization of filter cake and wastewater need financial support for transportation, but there is no financial analysis that will useful for the farmers.

Response        Thank you for the comment. The fermentation is used for ethanol production, which is not involved with waste utilization under the scope of this study. Reviewer may want to request the anaerobic digestion technology for vinesse treatment. We did add the list of available technology for the anaerobic digestion of vinesse. We also show the COD removal efficiencies of the anaerobic digestion through methanation.

Line 322-328:

Methanation through anaerobic digestion can reduce COD by 58–80%, but the effluent still contains a high COD of about 15,000–46,000 mg L ­1 (Moraes et al., 2015). In addition, the effluent is darker by melanoidin intensification (Moraes et al., 2015). The available technology for the anaerobic digestion of vinesse include up-flow sludge blanket, anaerobic fluidized bed reactor, anaerobic horizontal-flow immobilized biomass, and continuous stirred-tank reactor (Júnior et al., 2016; Moraes et al., 2015).

The financial analysis is important, but unfortunately, we did not include the part on technical assessment in this manuscript. We provide the challenge and recommendations for further studies on the financial feasibility of waste transportation. We hope the researchers and policymakers find it essential to driving the BCG.

  1. The analytical data were less presented, and mostly not match with the experimental objective. For example, Fig 1. Just describe the color, Fig 3 and 8 just the maps and not related to BCG, Fig 4. is not clearly explained, Fig. 5 impact of drought and flood to sugarcane farmer.

Response        Our objectives are presented in the last paragraph of the introduction. The goals are 1) to assess the current situation and feasibility of applying the byproducts from sugar mills and distilleries back into the sugar plantations for irrigation, fertilization, and soil conditioning. 2) to discuss challenges and provide recommendations for sustainable waste utilization 

These findings can be beneficial for policymakers to make strategies leading toward a successful BCG economy in Thailand. 

We extract key information from our results and present it in graphics and tables.

  • Figure 1 is the color of wastewater and vinasse. Since the color is the main problem and is hard to remove from the vinasse, the readers might want to see it. Nonetheless, this figure has been removed from the current version. Only description in the text remains.
  • Figures 3 and 8 are the results of the farmer interviews. It might not directly relate to the BCG. We show spatial distributions of the yield (Figure 3) and water resources (Figure 8) so that the readers know the differences in this information across the geographical areas. We also discuss the geographical factors regarding the information.
  • Figure 4 is the heat map, showing the dependences of soil properties and type of fertilization. The discussion of Figure 4 is in section 3.3. Applications of the byproducts and wastes as soil conditioner. 
  • Figure 5 is from farmers’ interviews. The graphic can imply the farmers' vulnerability to the drought and flood, intensified by climate change. Therefore, industrial wastewater can more or less mitigate the impact.

  1. The authors suggested that the byproduct, such as vinasse and filter cake have potential to produce high cane yield, but there is no data of sugarcane yield affected by application of the by product.

Response We acknowledge your comment. Based on our objectives to assess the current situation and feasibility of applying the byproducts, discuss the challenges, and provide recommendations, our experimental design is focused on literature reviews, farmer interviews, and field sampling and surveys. We conducted no control experiments to precisely present how much the yield would be increased by applying the byproducts. Nonetheless, based on literature reviews, we offer good practices for using the byproducts as biofertilizers to improve yields. The details are in section 3.2.2 Reviews on using waste from sugar-ethanol industries as fertilizer.

Using the byproducts as the sole fertilizer is not feasible in maximizing yield for commercial crops. There are no single reviews that suggest doing that (see citation below). The yield increase is due to co-applications of the byproducts with other fertilizers to enhance overall performance and decrease chemical fertilizer usage. The increasing yields are, therefore, highly varied due to different application approaches.

Section 3.2.2. Line 317-319:

“Though the nutrient composition of typical vinasse could complied with the mineral fertilizer, but it will be necessary to incorporate some nutrients from mineral fertilizer (Carpanez et al., 2022).”

Section 3.2.2 line 341-348:

“Co-applications utilizing both the cake and inorganic fertilizer could remediate such limitations. Sharma et al. (2002) suggested the integrated use of filter cake and urea (1:1) enhanced the cane yield in calcareous soil. Studies from various sites in India, China, Sudan, Swaziland, Egypt, Cuba, and Brazil agree that using filter cake in combination with chemical fertilizers increases crop yield and saves on fertilizer costs (Prado et al., 2013). For those with a low C/N ratio of filter cake, mixing with sugar cane bagasse, which has a higher C/N ratio, might be beneficial for minimizing N losses (Prado et al., 2013).

The filter cake as a fertilizer can be composted with vinasse prior to field application (Patil et al., 2013). Alternatively, the filter cake can be mixed with cow dung and vermicomposted to increase nutrient availability (Prakash and Karmegam, 2010). Considering life cycle assessment, utilizing wastewater, ash, and filter cake from sugar mills as fertilizer could benefit the environment, especially by reducing natural resources consumption (Contreras et al., 2009). The waste to fertilizer conversion is estimated to reduce environmental impacts on human health by 1.2%, ecosystem quality by 0.28%, and minimize resource consumption by 31.2% (Contreras et al., 2009). Dotaniya et al. (2016) recommended composting filter cake with vinasse in the ratio of 1:2.5 to obtain a bio-compost enriched with ferrous sulfate (FeSO4), zinc sulfate (ZnSO4) and biofertilizer with N fixers and P solubilizers. This integrated use can mitigate the environmental problems from the direct application of vinasse on agricultural lands since the composting can provide higher soil phosphorus (P) content and lower salinity (Murillo et al., 1993).”

  1. The storyline of the manuscript does not clear, for example the arrangement of the section was not sequentially presented.

Response We appreciate your suggestions. We provide the bullets showing the main sections in the results and discussion. In this way, the reader should better understand the manuscript's structure.

Line 183-193:

“We divide results and discussion into five sections

  1. First, we analyze and review nutrient contents and substances of the by-products and wastes from the sugar mill and distillery industries.
  2. Second, we discuss possible gains and impacts through the use of these by-products and wastes as biofertilizers
  3. Third, we suggest possible application of these by-products and wastes as soil conditioner.
  4. Fourth, we discuss the needs and potential of using treated wastewater from the sugar mill and distillery industry for ferri-irrigation.
  5. Fifth, we also discuss challenges on utilizing wastes from the sugar mills and the ethanol-distillery industries through the case of Kanchanaburi, Thailand.”

  1. The manuscript should completed with current references rather than using references published in 2010-2013.

Response We acknowledge your comments on the current versions of the references. We try to find the updated references, with their published years 2014 or more updated. Nonetheless, there are some references which their findings could not be substituted or reemphasized by the updated ones. We consider them as the most current knowledge.

Here are more updated references newly introduced to this manuscript.

Carpanez, T. G., Moreira, V. R., Assis, I. R., & Amaral, M. C. S. (2022). Sugarcane vinasse as organo-mineral fertilizers feedstock: Opportunities and environmental risks. Science of The Total Environment832, 154998.

Júnior, A. D. N. F., Koyama, M. H., de Araújo Júnior, M. M., & Zaiat, M. (2016). Thermophilic anaerobic digestion of raw sugarcane vinasse. Renewable Energy89, 245-252.

Kushwaha, J. P. (2015). A review on sugar industry wastewater: sources, treatment technologies, and reuse. Desalination and Water Treatment53(2), 309-318.

Kuziemska, B., Wysokinski, A., & Klej, P. (2023). The Content, Uptake and Bioaccumulation Factor of Copper and Nickel in Grass Depending on Zinc Application and Organic Fertilization. Agriculture13(9), 1676.

Montiel-Rosales, A.; Montalvo-Romero, N.; García-Santamaría, L. E.; Sandoval-Herazo, L. C.; Bautista-Santos, H.; Fernández-Lambert, G. Post-industrial use of sugarcane ethanol vinasse: a systematic review. Sustainability 2022, 14(18), 11635. https://doi.org/10.3390/su141811635

Moraes, B. S., Zaiat, M., & Bonomi, A. (2015). Anaerobic digestion of vinasse from sugarcane ethanol production in Brazil: Challenges and perspectives. Renewable and Sustainable energy reviews44, 888-903.

Moran-Salazar, R. G., Sanchez-Lizarraga, A. L., Rodriguez-Campos, J., Davila-Vazquez, G., Marino-Marmolejo, E. N., Dendooven, L., & Contreras-Ramos, S. M. (2016). Utilization of vinasses as soil amendment: consequences and perspectives. SpringerPlus5(1), 1007.

Sánchez-Marañón, M., Romero-Freire, A., & Martín-Peinado, F. J. (2015). Soil-color changes by sulfuricization induced from a pyritic surface sediment. Catena135, 173-183.

Santos, F.; Eichler, P.; Machado, G.; De Mattia, J.; De Souza, G. By-products of the sugarcane industry. In Sugarcane biorefinery, technology and perspectives 2020, (pp. 21-48). Academic Press. https://doi.org/10.1016/B978-0-12-814236-3.00002-0

Silalertruksa, T.; Wirodcharuskul, C.; Gheewala, S. H. Environmental Sustainability of Waste Circulation Models for Sugarcane Biorefinery System in Thailand. Energies 2022, 15(24), 9515. https://doi.org/10.3390/en15249515

Sousa, R. M. O., Amaral, C., Fernandes, J. M., Fraga, I., Semitela, S., Braga, F., ... & Sampaio, A. (2019). Hazardous impact of vinasse from distilled winemaking by-products in terrestrial plants and aquatic organisms. Ecotoxicology and Environmental Safety183, 109493.

Wongkoon, T., Boonlue, S., & Riddech, N. (2014). Effect of compost made from filter cake and distillery slop on sugarcane growth. KKU Res J.19(Supplement issue), 250-255.

Yin, J., Deng, C. B., Wang, X. F., Chen, G. L., Mihucz, V. G., Xu, G. P., & Deng, Q. C. (2019). Effects of long-term application of vinasse on physicochemical properties, heavy metals content and microbial diversity in sugarcane field soil. Sugar Tech21, 62-70.

Reviewer 4 Report

Sustainability

Manuscript ID: sustainability-2630746

The Challenge of Waste-to-Foods in the Sugar-Distillery Industry in Driving Bio-Circular Green Economy

The authors have investigated the potential of converting the sugar-distillery industry wastes into value-added bioproducts. The work could be of general interest. The following comments should help further improve the quality of the work:

1-The title does not effectively represent the content of the manuscript and should be improved.
2-Please add one more keyword (up to 6 is allowed). Metadata, including keywords, are important in terms of the searchability of the manuscript if published.
3-Please include a Table of Abbreviations/Nomenclatures.
4-Abstract should be improved by including the main numerical findings/conclusions of the study.
5-The novelty/originality of the paper should be more effectively established.
6-Please make sure all the units will be presented in compliance with the SI System. For instance, please use "L" instead of "liter" or “l”, “d” instead of “day”, etc. This comment applied to Figures/
Tables, too, where applicable.
7-Latest trends in valorizing sugar industries` wastes into value-added products should be included and discussed to enhance the timeliness of the present work. Here`s an example: “A critical review of multiple alternative pathways for the production of a high-value bioproduct from sugarcane mill byproducts: the case of adipic acid”, which, if found useful by the authors, can be used.
8-In all panel figures, such as Figure 1, please describe the panels (i.e., a, b, c, d) in the respective captions and not in the figure.
9-More attention should be paid to punctuation marks. On some occasions, they are missing. For instance, “As shown in Table 3, higher yields were found in sandy soil, loamy soil, and clayey soil, all with low amounts of fertilizer, at <312.5 kg ha-1 yr-1, 312.5 to 625 kg ha-1 and 625 to 937.5 kg ha-1 respectively.”; a comma is missing and it should be changed to “As shown in Table 3, higher yields were found in sandy soil, loamy soil, and clayey soil, all with low amounts of fertilizer, at <312.5 kg ha-1 yr-1, 312.5 to 625 kg ha-1 and 625 to 937.5 kg ha-1, respectively.”
10-The quality of Figure 2 is low and should be enhanced graphically. Please present the figure in color, too.
11-The organization and structure of Table 2 are confusing and should be substantially improved.
12-The quality of Figure 5 is low and should be enhanced graphically. Please present the figure in color, too.
13-Please do not repeat units where unnecessary, like in “7.73 mm–144.8 mm” which could be presented as “7.73–144.8 mm”.
14-The quality of Figure 6 is low and should be enhanced graphically. Please present the figure in color, too.
15-Moreover, Figure 6 is a panel figure, and hence, the panels should be designated by letters (a, b, c, etc.), and then the letter should be explained in the figure caption instead of explaining them in the figure.
16-The same comments raised above apply to Figure 7.
17-Some headings are too lengthy and should be made concise. See “3.4.4. Perception of farmers on the treated wastewater quality from the Thai sugar mills and its impacts”, for instance.
18-The same comments raised above apply to Figure 9, which is also a panel figure.
19-Overall, the quality of the figures and their presentation should be substantially improved. All figures should be graphically enhanced and should be synched from the presentation perspective.
20-
Future studies should investigate the sustainability features of the findings presented using advanced sustainability assessment tools, including life cycle assessment, exergy, etc., as elaborated in recent work such as “The role of sustainability assessment tools in realizing bioenergy and bioproduct systems”, “Life cycle assessment for sustainability assessment of biofuels and bioproducts”, etc. Please briefly discuss this future research need using works such as the examples provided, but not necessarily limited to them, and highlight the importance of such additional assessments to direct future studies.
21-Please include and discuss the limitations of the present study as well.
22-The practical implications of the present study should be included as well.
23-Please change "4. Conclusions" to "4. Conclusions and prospects". Accordingly, please elaborate on the future research needs in this domain.
24-
Please add DOIs for the references.

Some language polishing is advisable.

Round 2

Reviewer 3 Report

Thank you for revising the manuscript, and please revise the manuscript as follows:

1. Remove Fig 1 and place it as a supplement data 

2. Please complete the caption of Table 2 and Fig 2, Fig 3, Fig 4, and Fig 5, so it is easier to understand. 

3. Please revise Table 3 to become more understanable, for example place the months as the table section and growth stage as the column

The English need to be repared by English editing 

Reviewer 4 Report

Sustainability
Manuscript ID: sustainability-2630746
The Challenge of Waste-to-Foods in the Sugar-Distillery Industry in Driving Bio-Circular Green Economy

The manuscript seems to have been partially improved, but some revisions are still needed before it can be considered for publication.

1-As previously commented “The title does not effectively represent the content of the manuscript and should be improved.” Despite the authors` explanation, this comments still stands. Please improve the title as per the original comment above.
2-As previously commented “
Please include a Table of Abbreviations/Nomenclatures.” This has been done, but the table should be located before the Introduction section and not in the middle of the manuscript. Moreover, no need to number this table. Accordingly, please update the table numbers throughout the manuscript.
3-
As previously commented “Abstract should be improved by including the main numerical findings/conclusions of the study.” This has not been done effectively.
4-Please use “tonne” instead on “ton”.
5-
As previously commented “Please make sure all the units will be presented in compliance with the SI System. For instance, please use "L" instead of "liter" or “l”, “d” instead of “day”, etc. This comment applied to Figures/Tables too, where applicable.” This has not been done effectively. Here`s one example “Molasses, with currrent production of ~1.9 M L day-1”. This comment has not been effectively observed with regard to tables and figures either. See Table 2, for instance.
6-
As previously commented “Latest trends in valorizing sugar industries` wastes into value-added products should be included and discussed to enhance the timeliness of the present work. Here`s an example “A critical review of multiple alternative pathways for the production of a high-value bioproduct from sugarcane mill byproducts: the case of adipic acid”, which if found useful by the authors, can be used.” This has not been done effectively. Please refer to the original comment above and improve the manuscript accordingly.
7-Please avoid the usage of first-person pronouns (e.g., we and our).
8-There are typos throughout the manuscript. See the spelling of “current” in “Molasses, with currrent production of ~1.9 M L day-1” as an example.
9-The explanation provided in the middle of Figure 1 is not proper. Please explain figures in their captions and not in the figure. The units in the caption have not entirely been presented based on the SI system, see “year”.
10-As previously commented “The organization and structure of Table 2 is confusing, and should be substantially improved.” The table has been improved but still needs more improvements. Please align-left the content instead of aligning them at the center.
11-The quality of Figure 4 should be enhanced. Please present the figure in color.
12-As previously commented “The quality of Figure 5 is low and should be enhanced graphically. Please present the figure in color too.” This has not been done effectively. The figure is still blurred and its resolution should be enhanced.
13-As previously commented “The quality of Figure 6 is low and should be enhanced graphically. Please present the figure in color too.” This has not been done effectively.
14-The quality of Figure 8 should be enhanced. Please present the figure in color.
15-As previously commented “
Future studies should investigate the sustainability features of the findings presented using advanced sustainability assessment tools, including life cycle assessment, exergy, etc. as elaborated in recent work such as “The role of sustainability assessment tools in realizing bioenergy and bioproduct systems”, “Life cycle assessment for sustainability assessment of biofuels and bioproducts”, etc. Please briefly discuss this future research need using works such as the examples provided, but not necessarily limited to them, and highlight the importance of such additional assessments to direct future studies.” This has not been done effectively. Please refer to the original comment above and improve the manuscript accordingly.
16-As previously commented “Please include and discuss the limitations of the present study as well.” This has not been done effectively.
17-As previously commented “The practical implications of the present study should be included as well.” This has not been done effectively.

There are typos throughout the manuscript.

Round 3

Reviewer 4 Report

The manuscript can be accepted for publication.